# Verifying the relationships among the variabilities of summer rainfall extremes over Japan in the d4PDF climate ensemble, Pacific sea surface temperature, and monsoon activity

Shao-Yi Lee[1], Sicheng He[1], Tetsuya Takemi[1]

[1]Disaster Prevention Research Institute, Kyoto University, Uji, Kyoto 611-0011, Japan

*Correspondence to*: Tetsuya Takemi (takemi@storm.dpri.kyoto-u.ac.jp)

**Abstract.**

Upper 99th percentile hourly, 90th percentile daily, and 90th percentile pentad rainfall was calculated over the four large Japanese islands for June–July every year in the 1952–2010 period, using 10 ensemble members of the 5 km resolution
d4PDF (database for Policy Decision-making for Future climate changes) climate ensemble, and 126 rain-gauges. The HDBSCAN (Hierarchical Density-Based Spatial Clustering of Applications with Noise) algorithm was used to cluster rain-gauges and d4PDF grid-points, based a multi-frequency metric. Six analysis regions were identified based on rain-gauge clusters. Spearman correlation was calculated between cluster rainfall extremes and standardised scores of five modes from the Rotated Extended Principal Component Analysis of Pacific SST (Sea Surface Temperature) anomalies. By order of
explained variance for the analysis period, these modes represent El Niño-Southern Oscillation (ENSO) growth, ENSO decay, warming trend mixed with climate variability (Trend+), non-canonical ENSO, and other Pacific Decadal Variability (PDV). Rain-gauges showed field significant correlation between hourly extremes and Trend+, and between daily extremes and PDV. d4PDF showed excessive spatially widespread correlation and anti-correlation with ENSO decay and non-canonical ENSO, respectively. Rainfall extremes were better related to indices describing the regional monsoon jet's
position and water vapour flux, for both rain-gauges and d4PDF. Modulation of the regional monsoon by the Pacific SST modes was used to explain the relationship between rainfall extremes and Pacific SST modes. The differences between rain-gauges and d4PDF may be caused by slower monsoon development in d4PDF.

## 1 Introduction

Extreme rainfall events in recent summers have prompted the identification of regional conditions favorable to such developments (Harada et al. 2020, Yokoyama et al. 2020, Nayak and Takemi 2021, Naka and Takemi 2023). There is concern that climate change is increasing the strength of such events (Nayak and Takemi, 2019; Kamae et al., 2021; Mori et al., 2021), and if so, there is interest in what spatio-temporal scales of rainfall are impacted (Fujibe 2016, Unuma and Takemi 2021). The authors wish to evaluate the changes of monsoon rainfall extremes (if any) in the future warmer climate; for this purpose, a standard procedure of calculating climate projections begins by comparing and verifying historical or control climate simulation with observations. The possible dependence of such extremes on internal climate variability complicates the calculation of climate projections since climate variability itself may change in the future climate. Hence, in this study we evaluated the relationships between the variabilities of summer rainfall extremes over western Japan, monsoon activity, and Pacific sea surface temperature (SST).

Many hemispheric-scale modes of climate variability may interact with the regional- and global-scale signals over Japan; however, since Japan is an island located at the edge of the Pacific Ocean, it seems highly possible that the modes of Pacific SST have impacts on extreme rainfall over Japan. The dominant mode of Pacific variability is the El Niño-Southern Oscillation (ENSO). Although ENSO takes place in the tropical Pacific, it influences SSTs in the mid-latitudes and even globally in the form of the "Pacific-North America pattern" (Horel and Wallace 1981, Hoskins and Karoly 1981; see Alexander et al. 2002 for a literature review).

Over East Asia, the development of an El Niño during winter influences the following summer through the "Anomalous Philippine Sea Anti-cyclone", persistent anti-cyclonic wind anomalies over the western North Pacific to the east of the Philippines which result in a wetter monsoon (e.g. Wang et al. 2000). The persistence of the El Niño signal has been explained by the "Indo-western Pacific Ocean Capacitor", or teleconnections into and returning from the Indian Ocean (Kosaka et al. 2013, Xie et al. 2016), which excite a series of alternating high-and-low pressure, temperature and convection across the region termed as the "Pacific-Japan" (P-J) pattern (Nitta, 1987).

While the El Niño-associated anomalous anti-cyclone is one lobe of the of the P-J pattern, the P-J pattern is excited not only by ENSO. A positive Indian Ocean Dipole (IOD) occurring by itself without an El Niño would also have a teleconnection to the Pacific. An extremely warm Indian Ocean since 2019 likely contributed to the extreme rainfall event of 2020 (Takaya et al. 2020), in conjunction with other exacerbating factors like the Madden-Julian Oscillation (Zhang et al. 2021). The P-J pattern can also be excited by Rossby wave breaking (Takemura and Mukougawa 2022), which increases during Niña-like conditions in summer (Takemura et al. 2020). Such results suggest that it will be difficult to find clean and robust relationships between ENSO phase and rainfall over Japan.

An early study using data for a 30-year period (1951–1980) found no evident relationship between the Baiu (monsoon front) rainfall and the Southern Oscillation Index (SOI) of the season (Ninomiya and Mizuno, 1987). A study using another 30-year (1963–1992) period data found that the relationship changed from Baiu rainfall lagging the Nino3 index during 1963–1977, to leading Nino3 during 1978–1992 (Tanaka, 1997). A study of rainfall in Fukuoka (in Kyushu) for about a century (1890–2000) using categorised SOI found heavy rainfall to lag the "Strong La Niña" category (Kawamura et al. 2001, Jin et al. 2005). Yet another study for 54 years (1958–2011) found that weather patterns associated with heavy rainfall over Japan occurred more frequently when the Nino3.4 index of the season was high (Ohba et al. 2015). Such examples suggest that the relationship between ENSO and rainfall over Japan may depend on the period of data or even the location being studied. ENSO has a period of about 5–7 years, so even a 100-year period would contain only about 20 cycles. A 400-year study using paleoclimate proxies concluded that the strength and even sign of the relationship varied over time, and proposed that the Pacific Decadal Oscillation modulated the relationship (Sakashita et al. 2016). It appears that scientific literature is far from unanimous regarding the relationship between ENSO and rainfall variability over Japan.

Since we were unable to draw satisfactory conclusions from those past studies, we decided to investigate the relationships between Pacific climate variability and extreme rainfall variability in the monsoon season over Japan in observations first, before applying the same methodology to climate models. The methodology was based on two considerations:

Firstly, although the SOI or Niño indices indicate the ENSO-related state of the atmosphere and ocean respectively, ENSO phases are not instantaneous states but processes that develop and decay over the course of two years, even when SOI/Niño indices may have similar values before and after the ENSO peak. Clearer relationships may emerge between rainfall and an indicator measuring the direction of ENSO progress.

Secondly, the value of SOI/Niño indices may reflect not only the ENSO state, but also the states of other modes, such as the above-mentioned Pacific Decadal Oscillation. This study will directly decompose Pacific temperature anomalies through rotated extended Principal Component Analysis (rePCA) and correlate between each mode and extreme rainfall, instead of using the SOI/Niño indices.

The structure of this paper is organized as follows: Section 2 describes the data and methods used. Section 3 describes the results from the of Pacific SST, rainfall clustering, and correlation of SST modes with clustered rainfall extremes. In Section 4, we discuss how the SST modes may act on rainfall through monsoon activity. Finally, Section 5 summarises the findings of the study.

## 2 Data and Methods

### 2.1 Data and tools

#### 2.1.1 Rainfall data

The database for Policy Decision-making for Future climate changes (d4PDF) is a collection of ensemble climate simulations covering the 1951–2010 period (Mizuta et al. 2017, Ishii and Mori 2020). A 100-member historical-warming (HPB) climate ensemble has been simulated by the Meteorological Research Institute (MRI) Atmospheric General Circulation Model (AGCM) version 3.2 (MRI-AGCM3.2; Mizuta et al. 2012), which provided rainfall at 60 km spatial and daily temporal resolution. Twelve members have been downscaled to 5 km spatial and hourly temporal resolution (Kawase
et al., 2023). "d4PDF" in this study will refer to the downscaled data; the AGCM will termed "d4PDF-AGCM". Ensemble mean rainfall extremes from 10 members of the 5 km d4PDF was compared against rain-gauge rainfall extremes from 126 meteorological stations in Japan. The analysis period was 1952–2010, or 59 years.

Rainfall during June and July was analysed. This is the season when the monsoon (Baiu) front passes over the study domain.
An analysis of typhoon frequency was carried out for each month using the International Best Track Archive for Climate Stewardship (IBTrACS; Knapp et al., 2018) for observations, and Webb et al. (2019) for d4PDF. The number of typhoons was found to increase substantially in August. Evaluating typhoon changes in the future climate is not trivial (e.g. Mori and Takemi, 2016 for a review), so the authors preferred to exclude typhoon-associated rainfall as much as possible, even though the typical calendar "summer" includes August. From a phenomenon perspective, August rainfall is climatologically quite
distinct from June–July rainfall over Japan. The 99th percentile hourly, 90th percentile daily, and 90th percentile running-pentad rainfall values in June–July of every year were calculated. Percentiles were calculated inclusive of timesteps with no rainfall.

#### 2.1.2 Sea Surface Temperature (SST) and climate mode indices

The historical (HPB) d4PDF-AGCM ensemble was driven by randomly perturbed SST from the Centennial Observation-Based Estimates of SST version 2 (COBE-SST2; Hirahara et al., 2014). The SST data was of 1° spatial resolution and monthly temporal resolution. Data over the Pacific in the domain of 100°E–60°W, 20°S–60°N was used. Data on the Atlantic side was removed. Data over the Sea of Okhotsk (135°E–160°E, 45°N–60°N) was removed, in order to reduce the influence of sea ice. The same 59-year period as the rainfall period (1952–2010) was used for analysis.
The Southern Oscillation Index (SOI; Ropelewski and Jones, 1987) and Interdecadal Pacific Oscillation Tripole Index (TPI; Henley et al., 2015) were provided by the United States of America National Oceanic and Atmospheric Administration (NOAA). The non-standardised version of the SOI was used, but standardisation has no impact on correlation. The filtered COBE-SST version of the TPI was used, with units of degrees Celsius. The North Pacific Gyre Oscillation (NPGO; di
Lorenzo et al., 2008) index was used, available at https://o3d.org/npgo.

#### 2.1.3 Other meteorological variables

Meteorological variables in the region of (125–130°E, 20–50°N) from the Japanese 55-year Reanalysis (JRA55; Kobayashi et al., 2015) and d4PDF-AGCM were used to evaluate the monsoon activity. The datasets were obtained at 1.25° spatial
resolution. 850 hPa daily mean temperature and specific humidity were used to calculate the monsoon front location. Zonal wind, meridional wind and specific humidity at pressure levels 1000, 925, 850, 700, 600 and 500 hPa were used to calculate water vapour flux. Monsoon activity in the simulations was evaluated using variables from d4PDF-AGCM using the same 10 ensemble members as the rainfall data. The period of 1958–2022 was used because JRA55 started from 1958.

 **2.1.4 Processing tools**

Coastlines and administrative boundaries were obtained from the Database of Global Administrative Areas version 4.1 (GADM; https://gadm.org). Principal Component Analysis and part of the data processing was carried out using Max-Planck Institute for Meteorology's Climate Data Operators software package (CDO; Schulzweida, 2022). Equivalent potential temperature was calculated using the NCAR Command Language (NCL, 2019). The rest of the data processing was carried out in Python 3, using python libraries NumPy (Harris et al., 2020), SciPy (Virtanen et al., 2020), pandas (McKinney, 2010) and xarray (Hoyer and Hamman, 2017). Clustering of rainfall and curve fitting was carried out using SciPy. Wavelet analysis was carried out using Pywavelets (Lee et al., 2019). Figures were prepared with the python libraries Matplotlib (Hunter, 2007), Cartopy (Met Office, 2010–2015), and the GNU Image Manipulation Program (GIMP, 2025).

**2.2 Method**

**2.2.1 Rotated Extended Principal Component Analysis (rePCA)**

The SST field from COBE-SST was cropped to the domain of 100°E–60°W, 20°S–60°N. SST anomalies were calculated by subtracting the long-term monthly mean of the analysis period from the monthly SST, at each grid-point. The anomalies were areal-weighted with the cosine of latitude. Principal Component Analysis was performed on the concatenation of five seasonal mean anomalies, centered on the JJA season of the indexed year, i.e. the sample of 1952 would consist of D[-1]JF (December 1951, January 1952, February 1952), MAM (March 1952, April 1952, May 1952), JJA (March 1952, April 1952, May 1952), SON (September 1952, October 1952, November 1952), and DJ[+1]F[+1] (December 1952, January 1953, February 1953) seasonal mean anomalies. This is known as extended Principal Component Analysis, or This is also a Multi Singular Spectrum Analysis with embedding dimension 5 and trimestral sampling. The anomalies were not standardised before PCA. Extended PCA has been used in past studies to obtain the time evolution patterns of Pacific SST anomaly modes (Weare and Nasstrom 1982, Guan and Nigam 2008).

After extended PCA, areal-weighting was removed. Standardised scores (scores) and loadings were calculated from the Principal Components (PC) and Empirical Orthogonal Functions (EOFs), i.e. each PC (EOF) was divided (multiplied) by the square root of its corresponding eigenvalue to give the score (loading), so that the strength of the loading indicates its contribution to the total variance of temperature anomalies. A varimax rotation was performed on the 29 top modes and the rotated modes reordered by their explained variance. Twenty-nine modes were used for rotation following Guan and Nigam (2008), who selected this number based on the number of modes being above "noise level".

COBE-SST (Ishii et al., 2005; Japan Meteorological Agency, 2006) was also compared with COBE-SST2, and produced similar climate modes except for the Trend-including mode (not shown). This mode was sensitive to the analysis time period, even for the same COBE-SST2 dataset, as will be described in Section 3.1 (Figs 1 and 2), where rePCA was carried out for the 1921–2022 and 1952–2022 periods to examine the robustness of the resultant modes. The rePCA results used for comparison with rainfall were based on rePCA of the 1952–2010 period, the same period as the rainfall.

### 2.2.2 Rainfall clustering

Upper percentile rainfall in June–July of every year was calculated individual rain-gauges, as well as individual grid-points
of d4PDF within a convex hull covering four large Japanese islands of Hokkaido (northern diamond-shaped island), Honshu (S-shaped large island), Shikoku (southern barbell-shaped island) and Kyushu (southwestern oval-shaped island). The last two days of May and the first two days of August were used in the calculation of running-pentads. The percentile calculation was inclusive of times with zero rainfall. For hourly rainfall, this was the 15th highest value of the June–July season. For daily and running-pentad rainfall, these were the sixth highest values. This produced time-series of 59 samples (years). For
d4PDF, percentiles were calculated for each individual ensemble member, then the ensemble mean was taken.

The hourly rainfall in June–July of every year at individual locations was also decomposed into different frequency components through projection into wavelet space, using stationary (non-downsampled discrete) wavelet transform based on the Haar wavelet with six decomposition levels. A prior examination of the hourly data using continuous wavelet
transformation on the rainfall indicated that most of the energy lay within frequencies resolved by six discrete decomposition levels. This produced six detail components of periods 2–4 hours (D1), 4–8 hours (D2), 8–16 hours (D3), 16 hours–1.3 days (D3), 1.3–2.7 days (D4), 2.7–5.3 days (D5), 5.3–10.7 days (D6), and one approximation component containing signals of periods longer than 10.7 days (A6). The number of samples each component remained the same since the input was not downsampled, so the 99th percentile of each June–July season was recorded for each component. Hence, the rainfall at each
location was reduced to a 59×7 (years × frequency components) matrix. Wavelet transformation (Torrence and Compo 1998) has been successfully used to analyse rainfall at the Matsuyama rain-gauge (Santos et al., 2001) and precipitation indices over Japan (Duan et al., 2015), although for monthly resolution.

The spatial points were clustered using the Hierarchical Density-Based Spatial Clustering of Applications with Noise
(HDBSCAN) function, SciPy's implementation of the hierarchical density-based clustering algorithm developed by Campello et al. (2013). This algorithm builds a hierarchical tree of the samples (points) by searching around the vicinity of each sample at increasing radius based on a distance metric ($D$), merging samples into increasingly larger clusters as they are found, then returning the most persistent clusters above a minimum cluster size ($c_{min}$) along the tree's depth. The tree-construction algorithm was user-set to brute-force ("brute"), i.e. the tree was explicitly constructed. The manner in which
clusters grew to fulfill the requirement of increasing $c_{min}$ depended on the minimum sample size, which was set to 1, i.e. even one sample that was significantly correlated with any member of the cluster would be added to the cluster, allowing the cluster to keep expanding until no more such samples could be found.

The distance metric between two points $i$ and $j$ was defined as $D = 1 - r_{ij}$, where the geographically-adjusted similarity
$r_{ij} \in [-1, 1]$. For each clustering of points based on their hourly, daily and running-pentad rainfall extremes,

$$r_{ij} = \begin{cases} -1 & \text{if} \quad R_{ij} \text{ is statistically significant and negative} \\ 0 & \text{if} \quad R_{ij} \text{ is not statistically significant} \\ \max(0, R_{ij} - s_{ij}) & \text{if} \quad R_{ij} \text{ is statistically significant and positive} \end{cases}$$

using the Spearman correlation $R_{ij}$ and geographical adjustment factor $s_{ij} = S_{ij}/S_{max}$. The Spearman correlation $R_{ij}$ between
the rainfall time series of points $i$ and $j$ was evaluated for statistical significance using a two-tail test at $\alpha=0.05$. i.e. significant if $|R_{ij}| > 0.256$ for the 59-sample (years) period of 1952–2010. The geographical adjustment $s_{ij}$ is the physical distance between the two points $S_{ij}$ normalised through division by a maximal distance $S_{max}$, which will be described later. This was metric is similar to clustering by F-madogram (Bernard et al., 2013) with geographical adjustment factor (Bracken et al., 2015), and has been successful applied to the Tibetan Plateau, an orographically complex region like the study domain
(Ma et al., 2020).

For clustering of points based on wavelet decomposition, there were multiple Spearman correlation coefficients $R_{ij}^k$ and their corresponding similarity values $r_{ij}^k$, for $k=0...n_k$ where $n_k=6$, representing the frequency components A6, D6, ..., D1. In this case, $r_{ij}$ was defined as

$$r_{ij} = \begin{cases} -1 & \text{if} \quad r_{ij}^k \text{ is negative for any } k \\ \max\left(0, \sum_k r_{ij}^k/n_k - s_{ij}\right) & \text{otherwise} \end{cases}$$

where $r_{ij}^k = \begin{cases} -1 & \text{if} \quad R_{ij}^k \text{ is statistically significant and negative} \\ 0 & \text{if} \quad R_{ij}^k \text{ is not statistically significant} \\ R_{ij}^k & \text{if} \quad R_{ij}^k \text{ is statistically significant and positive} \end{cases}$

When $n_k=1$, this reduces to the version used in clustering hourly, daily or running-pentad extremes. The maximum function in the calculation of $r_{ij}$ implements a lower bound of 0 in cases of significantly correlated distant points. At sufficiently large distances $r_{ij}$ would be attenuated to the case there were no statistically significant $R$'s, or $D=1$. Both cases were distinct from the case where significant anti-correlation was found, where the points should not be clustered and $D=2$. While the maximum function strictly necessary, it provided two advantages. Firstly, since $c_{min}$ was a user-provided parameter, a method of selecting an optimal $c_{min}$ was needed. When there was a step jump in $D$, we found that as the algorithm was iterated with increasing $c_{min}$, there was a change in the number of clusters when a certain $c_{min}$ threshold was reached. A plateau was reached when the number of clusters remained the same. For d4PDF, this took the form of a singular cluster consisting of almost all grid-points in the domain, and a few tiny clusters over the Japanese Alps. Such a configuration was not interesting, so the previous $c_{min}$ was selected. Secondly, by reducing $r$'s to only indicate desirable to cluster ($r>0$) or strongly undesirable to cluster ($r=-1$), most $r$'s were zero. This allowed for efficient storage of $r$'s as sparse matrices, which was particularly important for d4PDF with over 35 thousand points.

To speed up the calculation, $s_{ij}$ was approximated using degree distance for $S_{ij}$ and 20° for $S_{max}$, the approximate degree distance from southwest to northeast of the d4PDF study region. Two points with minimally significant $R$ of approximately 0.25 would only be completely attenuated when their geographical degree distance exceeded 5°, which was sufficiently generous when compared against the smaller geographical extent of clusters obtained from rain-gauges. In the case of d4PDF, a further threshold $S_{calc}$ was set, where $r_{ij}$ was calculated only for pairs $(i, j)$ where $S_{ij}<S_{calc}$. A token value of $S_{calc}=1°$ was set based on the size of the rain-gauge clusters, and the results were checked against results obtained from halving and doubling $S_{calc}$.

The results using the wavelet-based distance metric were compared against results from individually clustering three temporal resolutions (hourly, daily running-pentad) for rain-gauges, before being applied to d4PDF. The comparison method is described in the following section.

### 2.2.3 Selection of analysis regions

To combine the results of individually clustering hourly, daily and running-pentad extremes, we would like to group the rain-gauges into regions that persisted across the three temporal resolutions. The most suitable $c_{min}$ amongst the three temporal resolutions was commonly applied to all three resolutions. Rain-gauges belonging to the same set of clusters produced at three temporal resolutions were grouped together ("3Class"). To illustrate this, if (A, B, C, D, E, F, G) was an hourly cluster, (A, B, C) and (E, F, G) daily clusters, and (A, B, D) and (C, E, F, G) were pentad clusters, the 3Class regions would be (A, B) and (E, F, G). A group must contain at least two rain-gauges. The 3Class regions were compared with the clusters produced by the wavelet-based distance metric, and eight regions were finally selected for analysis. These are listed in Table 1 and the comparison process will be described in Section 3.2.

A time series of rainfall was calculated for each cluster or unclassified location, commonly referred to as "sets". The mean rainfall of the set was calculated at every timestep. The number of samples every year would only along the time dimension and remain the same between different sets for the calculation of percentile rainfall. Upper percentile rainfall for the set was then calculated every year from the set's rainfall time series.

When directly comparing regional values between rain-gauges and d4PDF, statistics were calculated from d4PDF clusters co-located with the rain-gauges of the region. A d4PDF cluster was considered as co-located with the rain-gauge if any grid-point belonging to that cluster lay within 0.05° of the rain-gauge location.

**Table 1. The 8 regions in Japan selected for analysis, and the rain-gauges in regions.**

| Region | Rain-gauges |
|--------|-------------|
| 1 | Akita, Aomori, Asahikawa, Esashi, Fukaura, Haboro, Hachinohe, Hakodate, Iwamizawa, Kitami-esashi, Kushiro, Kutchan, Miyako, Morioka, Muroran, Mutsu, Obihiro, Omu, Otaru, Rumoi, Sakata, Sapporo, Suttsu, Tomakomai, Urakawa, Wakkanai |
| 2 | Fukui, Fushigi, Kanazawa, Karuizawa, Matsumoto, Nagano, Suwa, Takata, Toyama, Takayama, Wajima |
| 3 | Ajiro, Chichibu, Choshi, Hamamatsu, Irago, Irozaki, Katsuura, Kofu, Kumagaya, Lake Kawaguchi, Mishima, Mito, Okunikko, Omaezaki, Onahama, Shizuoka, Tateno, Tokyo, Utsunomiya, Yokohama |
| 4 | Gifu, Hikone, Himeji, Iida, Kobe, Kyoto, Nara, Nagoya, Osaka, Sumoto, Tokushima, Tsuruga, Tsu, Ueno, Wakayama |
| 5 | Maizuru, Matsue, Sakai, Tottori, Toyooka, Yonago, |
| 6 | Akune, Fukuoka, Fukuyama, Hagi, Hamada, Hirado, Hiroshima, Hita, Hitoyoshi, Iizuka, Kumamoto, Kure, Matsuyama, Mount Unzen, Nagasaki, Okayama, Oita, Saga, Sasebo, Shimonoseki, Tadotsu, Takamatsu, Ushibuka, Uwajima |
| 7 | Aburatsu, Kagoshima, Makurazaki, Miyakonojo, Miyazaki, Tanegashima, Yakushima |
| 8 | Fukushima, Ishinomaki, Sendai, Shirakawa, Yamagata |

### 2.2.4 Evaluation of correlation coefficients (*R*)

In this study, Spearman correlation was used, which measured only monotonicity between two variables without need for linearity. The correlation coefficient $R$ was tested for statistical significance using a two-tailed test at $\alpha=0.05$. For the 59-

year period of 1952–2010, the threshold value is $|R| = 0.256$. Analysis of JRA55 used the 53-year period of 1958–2010, with threshold value $|R| = 0.270$. In the text, $R$ will be described as "perfect" if $|R| = 1.00$, "strong" if $1.00 > |R| \geq 0.75$, "moderate" if $0.75 > |R| \geq 0.50$, "weak" if $0.50 > |R| \geq 0.25$, and "no relation" otherwise.

When multiple significance tests are repeatedly performed, a number of false positives may occur. The number of positives must exceed a threshold for there to be statistical significance for the entire field, i.e. field significance. Rain-gauge $R$'s were tested for field significance following the methodology of Livezey and Chen (1983). If individual rain-gauges were considered independent samples, significance for at least 12 rain-gauges would be required for field significance. Since the method requires samples to be mutually independent, if sets produced by clustering were used as samples, significance for at

least three sets would be required for field significance.

### 2.2.5 Monsoon activity indices

The structure of the monsoon front during a heavy rainfall episode has been described by Matsumoto et al, (1971) as being characterised more so by a low-level jet than temperature gradient, with heavy rainfall brought about by mesoscale systems

associated with 1000 km wavelength disturbances. We defined indices that described the location and stability of the monsoon front, as well as the location, stability and strength of the low-level jet.

The daily location of the monsoon front was determined based on the methodology of Li et al. (2018). The 850 hPa equivalent potential temperature $\theta_e$ was calculated from daily mean temperature and specific humidity, based on equation 39

of Bolton (1981). At each discrete longitude value in the range 125–130 °E, the latitudinal rate of change of $\theta_e$, $(\partial \theta_e / \partial \varphi)_\lambda$, was then calculated. The monsoon belt was defined as present in the latitudes where $(\partial \theta_e / \partial \varphi)_\lambda \leq -0.04$ K km$^{-1}$, and the location of the front was defined as the mean latitude if more than one latitude point was found. If only one latitude point was found, then the front was defined as not present. After this, the presence of the front was checked at the two points directly west and east of each longitude point. If the front was absent at both sides, then the front was defined as not present

at the central point. The zonal mean of the front latitude was defined to be FLat.

The meridional and zonal components of the lower tropospheric water vapour flux (henceforth, just "WVflux") at each grid-point were calculated for each day by vertically integrating the product of daily mean specific humidity $q$ and vector wind $(u,v)$ through six pressure levels (1000, 925, 850, 700, 600, and 500 hPa). In a discrete form, the zonal WVflux is $\sum_i (q_i u_i +$

$q_{i+1}u_{i+1})$ $(p_i - p_{i+1})$ / 2g, for the $i$=1...5 pressure levels. The magnitude of the WVflux was calculated from its zonal and meridional components. At each discrete longitude value in the range 125–130 °E, the discrete latitude location with the maximum WVflux was located, then its zonal and meridional components were extracted at that location. The zonal means of the WVflux jet latitude and vector components were defined to be JLat and (QU, QV), respectively.

For each of the 61 days in June–July, the means of Flat, JLat, and (QU,QV) were taken over all the years of data, to obtain a 61-day time series. For d4PDF, this was done individually for each of the 10 ensemble members, to obtain $10\times61=610$ days. The time series QU and QV were fitted to sine functions $A sin[\pi(t\text{-}T)/L]+B$, with A, B, L and T as fit parameters. The time series JLat and FLat were fitted to logistic functions $A[1 + e^{-k(t\text{-}T)}]^{-1}+B$, with A, B, k and T as fit parameters. For d4PDF, the 610 days of all ensemble members were fitted together. The fits were defined to be the climatological seasonal cycles $(QU)_0$,

$(QV)_0$, $(JLat)_0$, and $(FLat)_0$. For any day of any year, the deviation from the climatological seasonal cycle was calculated to obtain anomalies $(QU)_a$, $(QV)_a$, $(JLat)_a$, and $(FLat)_a$. For example, $(QU)_a = QU - (QU)_0$.

For each year, the seasonal mean anomalies $\mu_{QU}$, $\mu_{QV}$, $\mu_{JLat}$, $\mu_{FLat}$, and seasonal variances $\sigma^2_{Q(U,V)}$, $\sigma^2_{JLat}$, $\sigma^2_{FLat}$ were calculated from the anomalies to obtain seven annual time indices. For d4PDF, the indices were calculated individually for

each member of the ensemble, then the ensemble mean was taken. For example, $\sigma^2_{Q(U,V)} = [ \sum_i (\sigma^2_{QU} + \sigma^2_{QV})_i ]/N$ for $i$=1...N ensemble members (N=10).

## 3 Results

### 3.1 Rotated Extended Principal Component (rePCA) Analysis of Pacific SST anomalies

rePCA was carried out on three time periods, 1921–2022 (102 years), 1952–2022 (71 years), and 1952–2010 (59 years). The top five modes appeared distinct from the remaining modes in terms of explained variance. For the 1921–2022 period, the top five modes accounted for 24%, 16%, 8%, 6%, and 5% of the variance. For the 1952–2022 period, they accounted for 28%,16%, 7%, 6%, and 4% of the variance. For the 1952–2010 period, they accounted for 26%, 16%, 9%, 5%, 7% of the variance, with the order of the fourth and fifth modes swapped to match the order from the other two periods. The remaining
modes each explained 3% or less of the total variance. The top five modes were similar to those from Guan and Nigam (2008), which had identified the first five modes of their rotated eigenvectors as ENSO+ (growth), ENSO- (decay), Trend, and Pan-Pacific Decadal Variability. and non-canonical ENSO. In this study, these modes will be termed canonical ENSO growth (ENSO+), canonical ENSO decay (ENSO-), Trend-containing (Trend+), Pacific Decadal Variability (PDV), and non-canonical ENSO (ENSO-NC). Figures 1 and 2 show the scores and loadings, respectively.

The scores for ENSO+, ENSO-, PDV, and ENSO-NC were almost identical where they overlapped between the three time periods. This was particularly so for the four ENSO and PDV modes, where the relationships were almost perfect ($R>0.95$) between time periods (Figs 1a, 1b, 1c, 1e). The loadings were similar between time periods as well (not shown). Hence, these four modes were robust irrespective of the time period used for rePCA. To evaluate the identity of the modes against
known modes of Pacific variability in the scientific literature, their scores were correlated against SOI, NPGO and TPI for the 1952–2010 period. The negative of SOI was used such that positive correlation with -SOI indicated better correspondence to canonical El Niño. The correlation between ENSO+ score and -SOI was strong ($R=0.78$; Fig. 1a), and the loading clearly showed a canonical El Niño developing over Eastern Tropical Pacific (Figs 2a-e). The correlation between ENSO- score and -SOI leading by one year was moderate ($R=0.70$; Fig. 1b), and the loading clearly showed a decaying
canonical El Niño (Figs 2f-j). However, the ENSO-NC had no relationship with -SOI, and only weak anti-correlation with -SOI leading by one year ($R=-0.25$; Fig. 1e). The score was strongly positive in 1983 and 1998, years containing the tail end of strong prolonged El Niño events. In addition, the loading showed the Niño-like warm SST anomalies over the Eastern Tropical Pacific from March to August (Fig 2z-D). This indicated the ENSO-NC mode describes the tail end of an atypically strong El Niño that extends into summer months. PDV score had weak relationships with both NPGO ($R=-0.32$) and TPI
($R=0.49$; Fig. 1d). The anti-correlation between TPI and PDV was the strongest relationship between TPI and the SST modes (ENSO+ 0.29, ENSO- -0.27, Trend+ -0.27, ENSO-NC -0.18). The correlation between NPGO and PDV was not the strongest relationship between NPGO and SST modes (ENSO+ -0.31, ENSO- -0.15, Trend+ 0.38, ENSO-NC 0.16).

The score and loading of the Trend+ mode was sensitive to the analysis period. As shown in Fig. 1c, the scores for the 1952–
2010 and 1952–2022 periods were strongly correlated ($R=0.95$), but both were different from the score for the 1921–2022 period (R=0.62 with the score for the 1952–2010 period). Although the scores of Trend+ overall increased in all three analysis periods, different interannual and decadal variability could be seen, which may contribute to the lower $R$'s. The spatial loading of Trend+ was not homogenous but showed seasonal and spatial variability, with cool anomalies over the tropical Pacific resembling the La Niña and warm anomalies over the northwest Pacific (Figs 2k-o). Although the Niña-like
pattern was consistent with some past studies, other studies have also reported a Niño-like pattern, with the direction of the pattern depending on the choice of data used (e.g. Vecchi et al. 2008, Lee et al. 2022). The loading from the 1921–2022 period showed stronger warm anomalies over Northwest Pacific, but weaker "horseshoe-shape" cool anomalies over East Pacific (Figs 2p-t). Hence, the spatial patterns in Fig. 1k-o may be residues from interannual and decadal variability. The actual global warming pattern associated with only the warming trend may be much more homogenous. The correlation
between NPGO and Trend+ was the strongest relationship between NPGO and SST modes ($R=0.38$), which supports this possibility. For this reason, the mode was termed "Trend-containing (other climate variability)" rather than just "Trend".

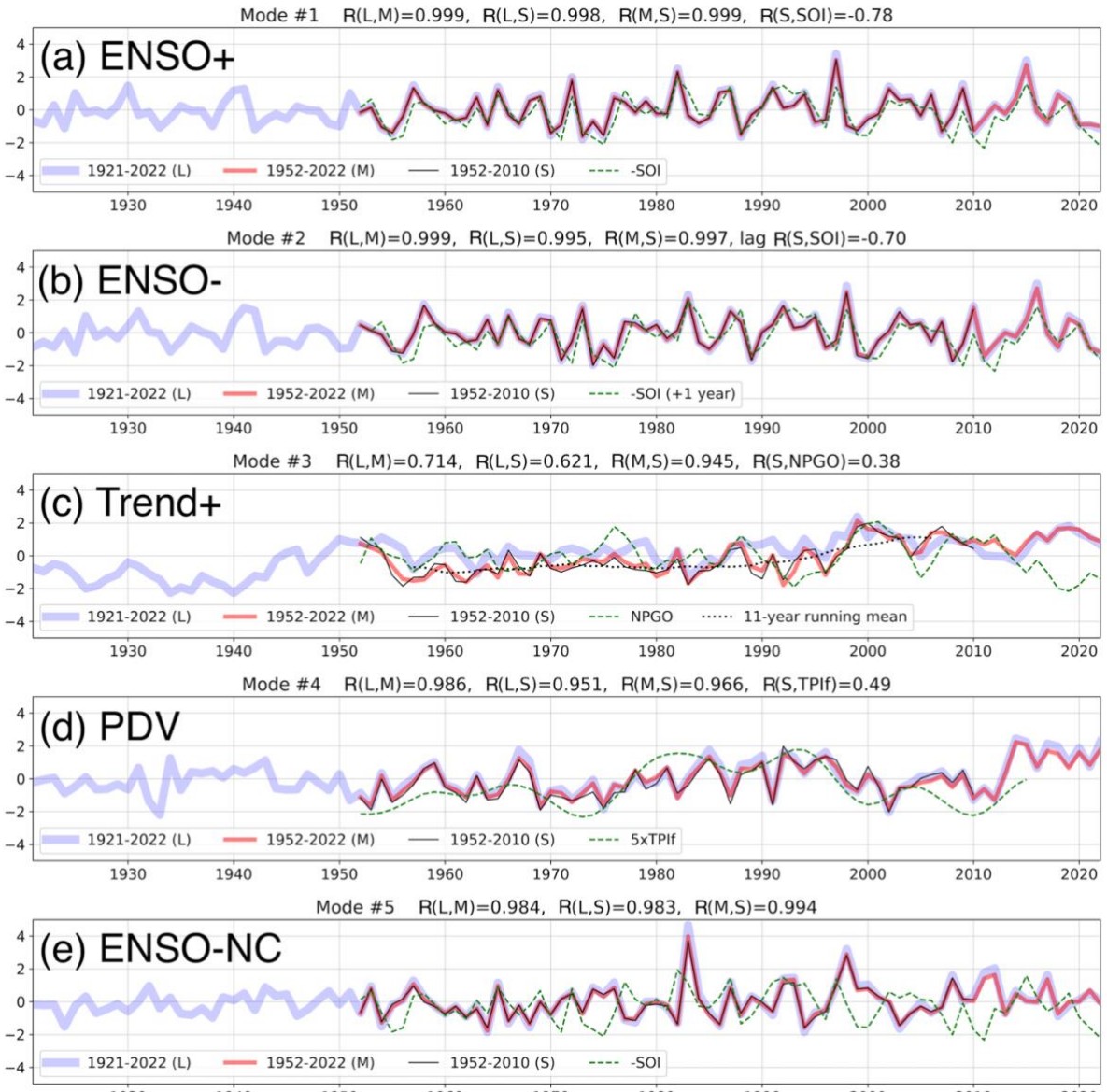

**Figure 1.** Standardised principal components (scores) of the first five modes from the rotated extended PCA of Pacific SST anomalies for the 1921–2022 (blue), 1952–2022 (red), and 1952–2010 (black) periods. (a) Canonical ENSO-growth mode "ENSO+"; (b) Canonical ENSO-decay mode "ENSO-"; (c) Trend-containing mode "Trend+"; (d) Pacific Decadal Variability mode "PDV"; (e) non-canonical ENSO "ENSO-NC". The black dotted line in panel (c) shows the 11-year running mean of the 1952–2010 score. The green dashed line in the panels shows (a) negative of the Southern Oscillation Index "-SOI"; (b) negative of the Southern Oscillation Index phased-shifted by 1 year "-SOI (+1 year)"; (c) North Pacific Gyre Oscillation index "NPGO"; (d) Interdecadal Pacific Oscillation Tripole Index filtered version "TPIf" (in °C scaled by a factor of 5); (e) negative of the Southern Oscillation Index "-SOI". The correlation coefficients between the scores from 1921–2022 "L", 1952–2022 "M", and 1952–2010 "S" for overlapping periods are shown on the top of the panels, as well as correlation coefficients between "S" and the green index for panels (a)-(d).

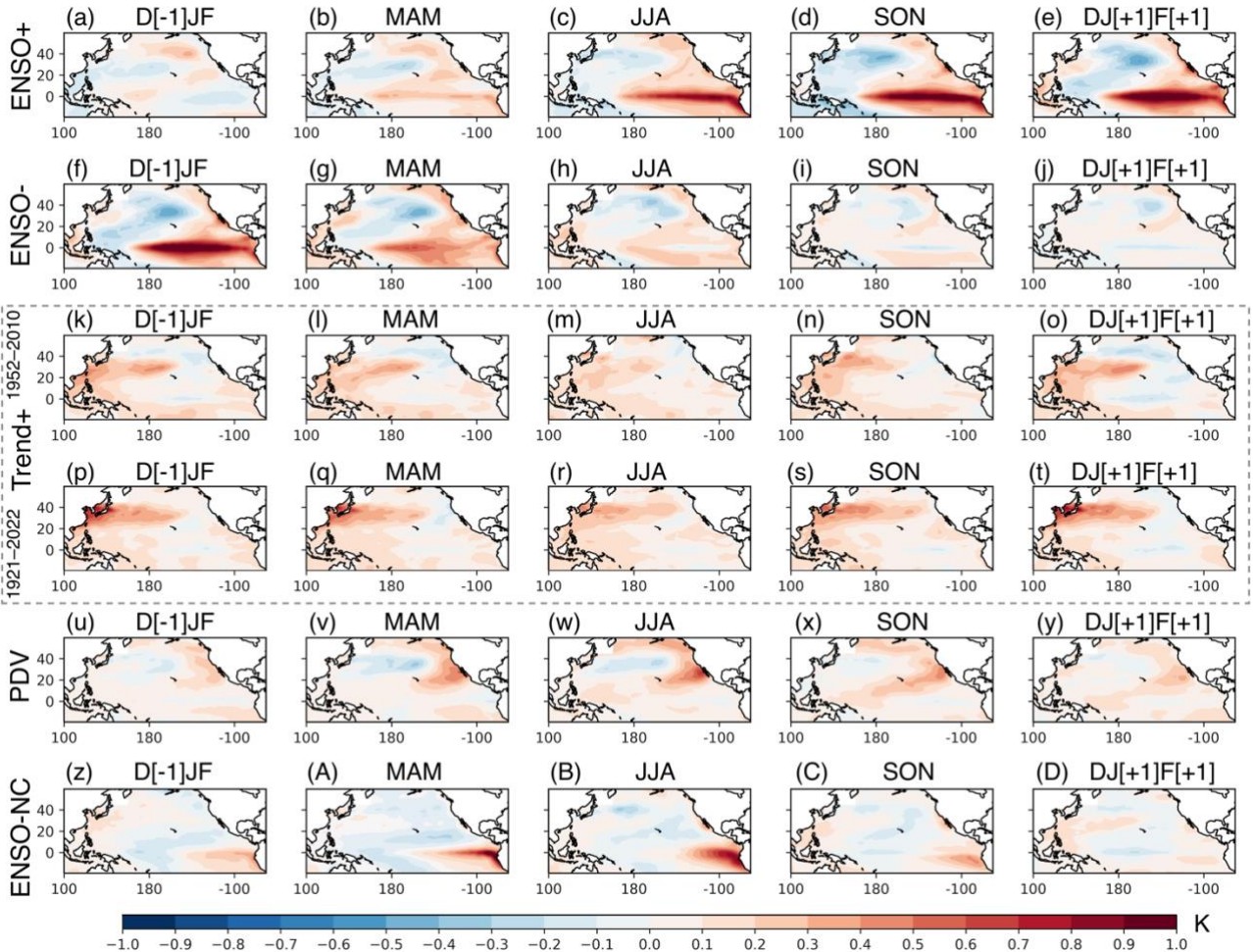

Figure 2. Loadings of the first five modes from the rotated extended PCA of Pacific SST anomalies for the 1952–2010 period, and only the Trend+ mode for the 1921–2022 period. Rows show the modes. Row 1 (a-e) ENSO+; row 2 (f-j) ENSO-; row 3 (k-o) Trend+; row 4 (p-t) Trend+ for 1921–2022; row 5 (u-y) PDV; row 6 (z-D) ENSO-NC. Columns show anomalies of five seasons centered on June-July-August "JJA".

## 3.2 Rainfall clustering

The results of individually clustering three temporal resolutions of rain-gauge rainfall are shown in Figs 3a–d. For hourly extremes, $c_{min}=4$ produced 12 clusters with 28 unclassified rain-gauges, but $c_{min}=5$ and $c_{min}=6$ produced 6 clusters with 19 unclassified rain-gauges (Fig. 3a). For daily extremes, $c_{min}=5$ produced eight clusters with 39 unclassified rain-gauges (Fig. 3c), but $c_{min}=6$ and $c_{min}=7$ produced three clusters with five unclassified rain-gauges (Fig. 3d). For running-pentad extremes, $c_{min}=4$ produced 14 clusters with 3 unclassified rain-gauges, but $c_{min}=5$ and $c_{min}=6$ produced 2 clusters with 6 unclassified rain-gauges (Fig. 3b). The three results were similar despite some minor differences in whether smaller clusters were merged into larger clusters or turned into unclassified rain-gauges due to being smaller than $c_{min}$.

There were three clusters in Hokkaido island, represented by clusters 3, 4, and unclassified rain-gauges on the southern coast in Fig. 3c. These may merge into one larger cluster (cluster 2 in Fig. 3b; cluster 3 in Fig. 3d). There were two clusters over the Tohoku region, represented by clusters 1 and 2 in Fig. 3c. The northern cluster may either merge northwards (Fig. 3a) or southwards (Fig. 3b) to form one larger cluster. The southern cluster usually merged southwards (cluster 1 in Fig. 3b and Fig. 3d). There was a large cluster in the Kanto region, represented by cluster 2 in Fig. 3a and cluster 6 in Fig. 3c. The size and boundary of this cluster varied particularly in terms of its western coastal side and northern mountainous side; North of it was a small cluster on the Sea of Japan coast (cluster 3 in Fig. 3a), while west of it was a cluster in the Kansai region (cluster 5 in Fig. 3a and Fig. 3c). There was a large cluster covering most of the Chugoku region and Kyushu island, represented by cluster 6 in Fig. 3a and cluster 8 in Fig. 3c. Finally, there was a small cluster at the southern tip of Kyushu that was quite robust persisting through multiple values of $c_{min}$, represented by cluster 4 in Fig. 3a and cluster 5 in Fig. 3c. The five clusters from Kanto to Kyushu may merge into one larger cluster (cluster 1 in Fig. 3b and Fig. 3d). Most rain-gauges along the Pacific coast of Shikoku island and Kinki were unclassified and only merged into this larger cluster as $c_{min}$ increased.

The optimal value $c_{min}=6$ was common between the three temporal resolutions. Fig. 3e shows "3Class" rain-gauge groups with the same combination of cluster labels from using $c_{min}=6$ on the three temporal resolutions, excluding 23 rain-gauges containing any unclassified label. Six 3Classes emerged, with the "5" label skipped for later comparison with Fig. 3f. The northernmost being a four-in-one large cluster consisting of the combination of the three Hokkaido clusters and the northern Tohoku cluster. The southern Tohoku cluster was excluded because it consisted of five rain-gauges, which became unclassified for hourly extremes when $c_{min}=6$ (Fig. 3a). Based on this consideration, the southern Tohoku cluster was manually added to create a total of seven 3Classes (red dashed contour labelled "8" in Fig. 3e).

Fig. 4 shows the clusters returned with different $c_{min}$ using the wavelet metric. Starting from $c_{min}=3$ produced 19 clusters and 23 unclassified rain-gauges (boxes a–s in Fig. 4). Increasing to $c_{min}=4$ produced 12 clusters with 22 unclassified rain-gauges; Five smaller clusters in Kyushu island and Chugoku region merged (boxes n–r in Fig. 4; cluster 6 in Fig. 3e), but the cluster at the southern tip of Kyushu (box m in Fig. 4; cluster 7 in Fig. 3e) and the cluster on the Sea of Japan coast (box s in Fig. 4; cluster 5 in Fig. 3e) would persisted until $c_{min}=11$. Increasing to $c_{min}=5$ produced nine clusters with 20 unclassified rain-gauges; three smaller clusters in the Kanto region merged into one (boxes i–k in Fig. 4; cluster 3 in Fig. 3e). A notable change took place at $c_{min}=6$ which produced 7 clusters and 18 unclassified rain-gauges. Hokkaido and northern Tohoku clusters merge into one cluster (boxes a–e in Fig. 4; cluster 1 in Fig. 3e). However, the southern Tohoku cluster of five rain-gauges became unclassified (box f in Fig. 4). The clusters remained largely unchanged until $c_{min}=11$, except that the Sea of Japan cluster of six rain-gauges also became unclassified at $c_{min}=7$. Hence, $c_{min}=6$ was optimal for producing a small number of large clusters, as shown in Fig. 3f. The southern Tohoku cluster was manually added to create a total of eight analysis regions and 12 unclassified rain-gauges (box f in Fig. 4; cluster 8 in Fig. 3f). The specific rain-gauges that make up these regions have been listed in Table 1.

The result of individually clustering three temporal resolutions and creating 3Classes from unique combinations of cluster labels and the result of the wavelet-based clustering were compared in Fig. 3e and Fig. 3f. The same optimal $c_{min}=6$ was found in both methods. The value of $c_{min}$ can be reduced to produce better geographical resolution, since most of the smaller clusters appeared robust between methods. On the other hand, increasing $c_{min}$ further would not produce interesting clusters from the perspective of regional study since the results would resemble Fig. 3b or Fig. 3d. The clusters produced by the two

methods were also highly similar. The most notable difference was that that 3Class 6 in Fig. 3e split into two wavelet-based clusters, clusters 5 and 6 in Fig. 3f. Minor differences were that clusters 1 and 2 were larger from the wavelet-based method than the 3Class method. The wavelet-based method is better than the 3Class method in that it included multiple temporal frequencies in one round of clustering, and produced a single clustering tree that was more easily interpretable.

The wavelet-based clustering method was next applied to d4PDF. When $c_{min}=8$, most points suddenly merged into one large cluster covering the entire study domain. Therefore, $c_{min}=7$ was used, and the result is shown in Fig. 5a. There were 1501 small clusters, which appeared excessively large. To make sense of this, three wavelet components were independently clustered. These were D1 corresponding to periods of 2–4 hours, D4 corresponding to periods of 1.3–2.7 days, and D6 corresponding to periods of 5.3–10.7 days. Fig. 5b shows the results of clustering D1 with the largest usable $c_{min}=7$, producing 898 classes. Clusters with similar (purple) labels were found along Kyushu and the Pacific coast of western Japan. Fig. 5c shows the results of clustering D4 with the largest usable $c_{min}=8$, producing 1165 clusters. The linear pattern along the Pacific coast had broken apart. In particular, cluster labels over Kyushu were split into western (purple) and eastern (orange–yellow) groups. Fig. 5d shows the results of clustering D6 with the largest usable $c_{min}=25$, producing 282 clusters. Clusters with similar (orange–yellow) labels now covered the whole of Kyushu, inland and the Sea of Japan coast of northeast Japan. Purple labels were found over the region corresponding to cluster 5 in Fig. 3e and the Pacific coast from Kanto to Hokkaido. The change between two coast-aligned patterns from high to low temporal frequencies likely contributed to the fragmentation into many clusters. Illustrating this with rain-gauges, $c_{min}=5$ produced six hourly, eight daily and two running-pentad clusters, but these combined into nine 3Classes (10 with the manual addition of the southern Tohoku cluster).

Another contributing factor was likely limitation by the finest denominator of clustering. The D6 component could be clustered with $c_{min}=25$ into 282 clusters of relatively large size, so the overall $c_{min}=7$ threshold was likely due to the D1 component with the same threshold. To test if larger clusters could be formed by considering more distant points as cluster candidates, D1 was re-clustered after doubling the calculation radius $S_{calc}$ from 1° to 2°, then from 2° to 4°. The results were unchanged, indicating that the cause was the lack of significantly correlated points inside the domain at this frequency range, rather than artificial limitation by $S_{calc}$. Even if clustering were limited to D6, the 282 D6 clusters was still very large compared to even the $c_{min}=3$ rain-gauge result of 19 clusters. The reason for the large number of clusters is not clear. It is possible that clustering methodology must be further adapted for use on the high-resolution ensemble simulations.

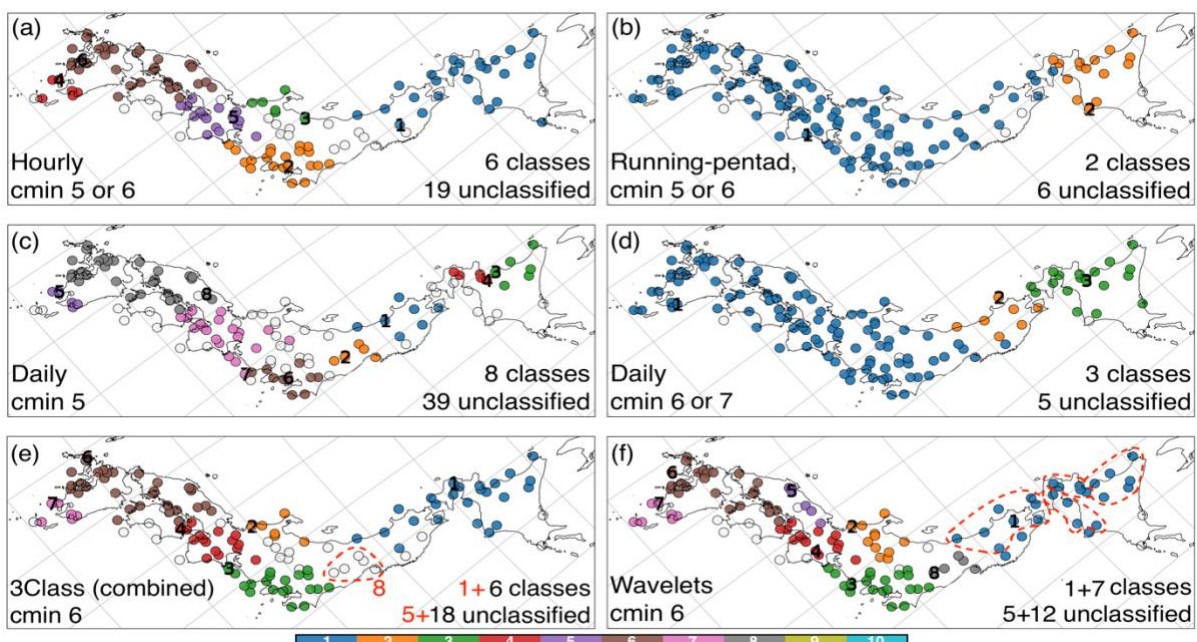

Figure 3. Rain-gauge clusters. Labels of clusters based on the time series of June–July (a) 99th percentile hourly rainfall using $c_{min}$=5 or $c_{min}$=6; (b) 90th percentile running-pentad rainfall using $c_{min}$=5 or $c_{min}$=6; and (c) 90th percentile daily rainfall using $c_{min}$=5; (d) 90th percentile daily rainfall using $c_{min}$=6. (e) Labels of station groups with the same combination of hourly, daily and running-pentad clusters using $c_{min}$=6 ("3Class"). The red dashed contour shows the manual addition of cluster 8. Cluster label 5 is missing in this panel. (f) Labels of clusters using the wavelet-based metric with $c_{min}$=6, including the manual addition of one $c_{min}$=5 cluster as cluster 8. Red dashed contours show three $c_{min}$=5 clusters that merged to form cluster 1 when $c_{min}$=6. The number of classes (referring to either clusters or 3Class) and unclassified rain-gauges are shown on the bottom right. Colours only reflect the cluster numbering and do not reflect any units or magnitudes. White markers show unclassified rain-gauges. Longitude-latitude gridlines are shown as grey lines; the map is a rotated projection.

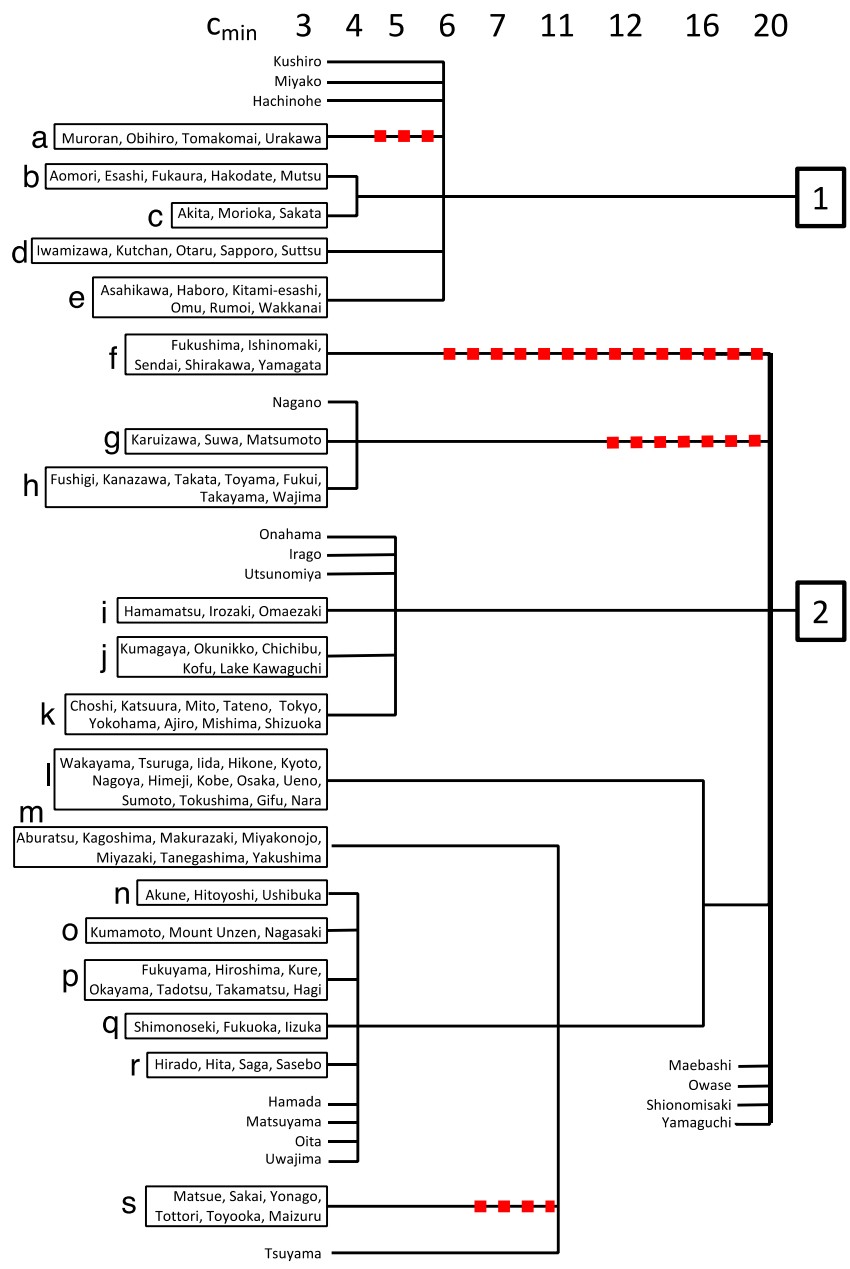


Figure 4. The combination of rain-gauges into clusters using the wavelet-based metric and different values of $c_{min}$, starting from 19 clusters when $c_{min}$=3. Red dashes indicate that the rain-gauges in the boxes became unclassified because the cluster size was smaller than $c_{min}$. These unclassified rain-gauges later merged back into larger clusters.

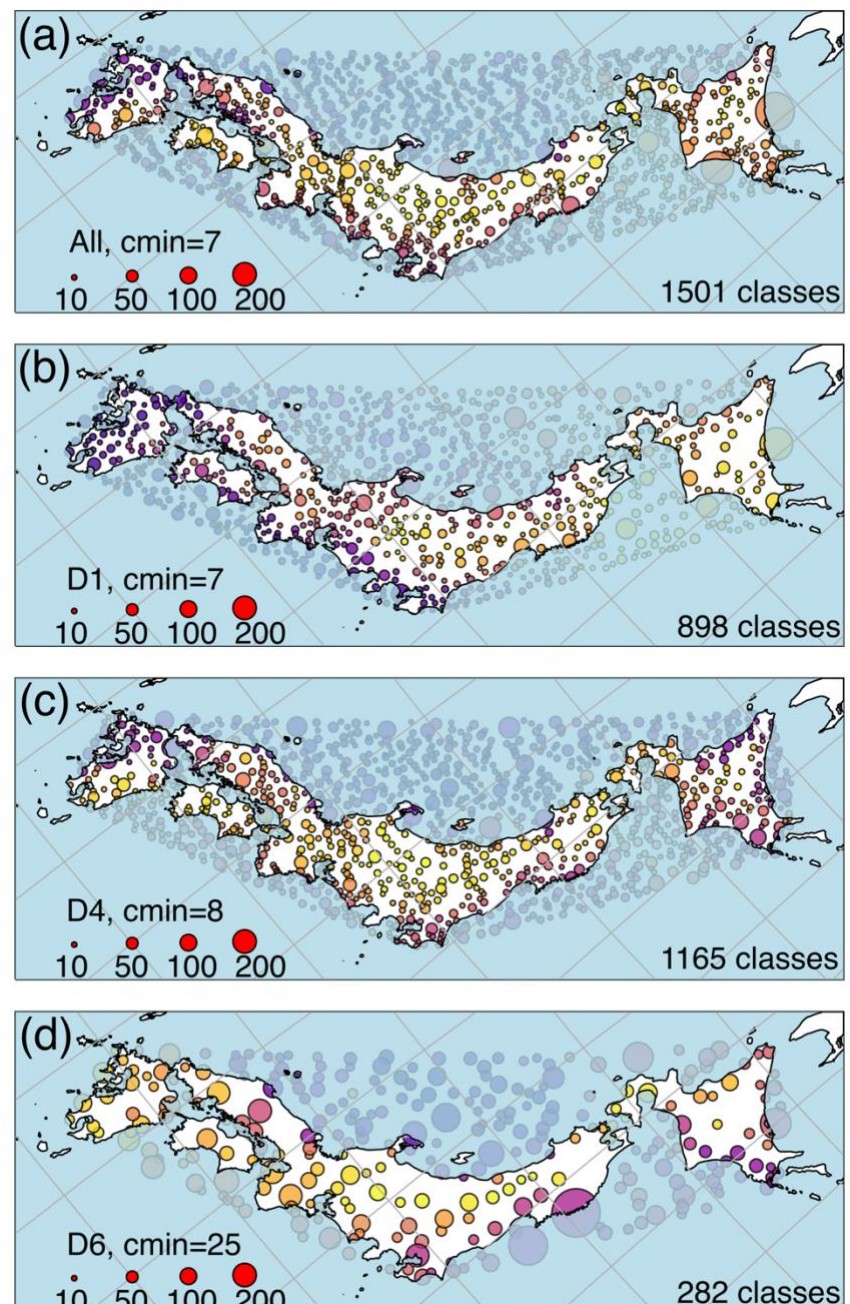

**Figure 5. d4PDF clusters.** Labels and sizes of clusters based on the time series of seasonal (a) 99th percentile hourly, (a) 90th percentile daily, and (a) 90th percentile running-pentad rainfall. Colours only reflect the cluster numbering and do not reflect any units or magnitudes. Circle marker sizes reflect cluster sizes (legend). A translucent blue mask was used to cover data within the convex hull used for clustering, but falling inside the ocean. Longitude-latitude gridlines are shown as grey lines; the map is a rotated projection.

### 3.3 Correlation Coefficients (*R*'s)

Before correlating rainfall extremes with the Pacific SST modes, the rainfall values to be used were compared between rain-gauges and d4PDF clusters co-located with the rain-gauges. Figure 6 shows the distributions of rainfall extremes for the eight analysis regions. The ranges of rain-gauges and d4PDF overlapped in all comparison cases, although the distribution
peaks of d4PDF were always of smaller value than that of the rain-gauge. The latter was not surprising since the model was of 5 km horizontal resolution, and we expect rainfall extremes to improve with model resolution in the future. Hourly and daily extremes were better simulated than running-pentad extremes; The quartile bars of rain-gauges and d4PDF overlapped except for Region 5. For running-pentad extremes, the quartile bars of rain-gauges and d4PDF overlapped only for Region 3 (Kanto). The difference in performance between daily and running-pentad resolutions suggests that rain-gauges experienced
more continuous rainfall in 5-day periods, but the model less so. d4PDF distributions of hourly extremes are bimodal or trimodal in most cases, but for rain-gauges this was only obviously so for Region 2 (central Sea of Japan coast). d4PDF performed the best over Region 3, and worst over Region 5 (western Sea of Japan coast). The good performance over Region 3 may be due to it being topographically flat. In contrast, Region 5 was a topographically complex region with steep topography near a coastline.

Figures 7–9 show the R's between rainfall extremes and scores of the Pacific SST modes, at three temporal resolutions. Figure 10 summarises the results by analysis region. The left columns of Figs 7–9 show R's for individual rain-gauges, which were at most weak. By individual rain-gauges, field significance was found for correlation with Trend+ for hourly extremes (12 significant rain-gauges), and anti-correlation with PDV for daily extremes (13 significant rain-gauges). Rain-
gauges correlated with Trend+ for hourly extremes were scattered over the study domain, but more were found on the east side of Kyushu island, and on eastern Japan near the Kanto region (Fig. 7c). Due to their scattered nature, only analysis region 7 (southern Kyushu) showed significant hourly relationship of $R$=0.31 (first bar of mode 3 in Fig. 10b). Region 3 (Kanto) showed hourly $R$=0.22 close to but not exceeding the significance threshold (first bar of mode 3 in Fig. 10f). Rain-gauges anti-correlated with PDV for daily extremes were less scattered, with more found on Shikoku island and Hokkaido
island (Fig. 8d). Most rain-gauges on Shikoku specifically Pacific-facing locations were unclassified by clustering. Region 1 (Hokkaido and Tohoku) showed significant daily relationship of $R$=-0.28 (second bar of mode 4 in Fig. 10g).

Besides these larger patterns, there were rain-gauges concentrated over northwestern Kyushu correlated with ENSO+ for daily extremes (Fig. 8a), over Kanto anti-correlated with ENSO+ for running-pentad extremes (Fig. 9a), over Hokkaido anti-
correlated with PDV for hourly extremes (Fig. 7b), over west Kyushu correlated with ENSO-NC for hourly and daily extremes (Figs 7e and 8e), and along the Pacific coast of eastern Japan with ENSO-NC for hourly extremes (Fig. 7e). However, these relationships were not seen for the analysis regions that contained these subsets of rain-gauges except for the last, reflected in Region 1 which showed significant hourly relationship of R=0.28 (first bar of mode 5 in Fig. 10g).

Besides the regional relationships described above, region 7 showed significant running-pentad relationship with ENSO+ of $R$=-0.27 (Fig. 10b), and region 1 showed significant daily relationships with ENSO-NC of $R$=0.30, respectively (Fig. 10g). Examination of Fig. 9a shows that the first relationship derives from only one rain-gauge in the cluster. Examination of Fig. 8e suggests that the second relationship to be believable since it derived for four rain-gauges, and was consistent with the hourly ENSO-NC relationship. Considering clusters, there was no domain-wide field significance with any SST mode.

The right columns of Figs 7–9 show R's for d4PDF clusters, which were at most weak, except for the relationship between hourly extremes and ENSO-NC where a few moderately anti-correlated clusters were seen over eastern Japan (Fig. 7j). The spatial correlation patterns in d4PDF were different from those in the rain-gauges. Correlation with ENSO+ was seen over Chugoku for hourly and running-pentad extremes (Figs 7f and 9f; Region 5 in Fig. 10c). This was not seen in the rain-gauges,
although this may be a spatial bias where observed positive correlation over northwest/west Kyushu was shifted northwestwards in d4PDF. Correlation with ENSO- was seen over western Kyushu and some places along the Sea of Japan coast. (Figs 7g, 8g, and 9g; Regions 6 and 7 in Figs 10a and 10b) This was excessively strong and not seen in the rain-gauges, so it can be considered as a strength bias. Correlation with Trend+ was seen over some parts of Hokkaido (Figs 7h, 8h, and 8h). This may be a spatial bias from the rain-gauges, where correlation was seen in regions south of Hokkaido but not in

Hokkaido. Correlation with PDV was seen over eastern Kyushu and southeast Hokkaido for daily and running-pentad extremes (Figs 8i and 9i). This was strength bias from the rain-gauges, which only showed anti-correlation. Correlation with ENSO-NC was seen over some parts of western Kyushu, while anti-correlation with ENSO-NC was seen over eastern Japan (Figs 7j, 8j, and 9j; Region 3 in Fig. 10f). The anti-correlation over eastern Japan was excessive and not seen in rain-gauges, resulting in a strength bias.

The relative performance of d4PDF for the different SST modes can be evaluated from Fig. 10. We required that the rain-gauge $R$ values (blue diamonds) fall within the range of d4PDF $R$ values (box plot), for at least two out of three temporal resolutions, to call the model performance "acceptable". For ENSO+ this was six of eight regions (6, 5, 2, 1 / 3, 8 listed along rows with top and bottom rows divided by a slash). For ENSO- this was three regions (/ 4, 3, 8). For Trend+ this was seven regions (6, 5, 2, 1 / 4, 8). For PDV this was three regions (5 / 3, 8). For ENSO-NC this was four regions (6, 5 / 7, 4). The best performance by region was for ENSO+ and Trend+, although in almost all cases acceptable performance meant reproducing the lack of relationship. the Trend+ mode was described in Section 3.1 as including both warming trend and well as interannual and decadal variability, so we would like to further investigate its relationship with rainfall extremes.

To better evaluate the relationships between rainfall extremes and warming alone, Trend+ scores were smoothed using a 11-year running mean (black dotted line in Fig. 1c). Figure 11 shows the correlations with this smoothed trend. Figure 12 summarises the results by analysis region, including also regional relationships with the NPGO index, TPI index, and SOI. The spatial patterns for the other three indices are not shown because the relationships were unremarkable and no better than that with the SST modes. In these figures, the threshold for significance was $|R| > 0.273$ due to the shorter time period after smoothing. Due to this high threshold, there was no domain-wide field significance by considering individual rain-gauges. Although Fig. 11a showed a similar number of weakly correlated rain-gauges as Fig. 7a, including one moderately correlated rain-gauge at southern Kyushu, only 10 rain-gauges were significantly correlated. Nevertheless, the spatial distribution of correlated rain-gauges over Kyushu and anti-correlation rain-gauges over the inland region further northeast was present at all three temporal resolutions (Figs 11a, 11b, and 11c). d4PDF showed a similar spatial pattern, except for the presence of correlated regions over Hokkaido that was not seen in rain-gauges (Figs 11d, 11e, 11f).

By analysis region, only Region 7 showed significant correlation for all three temporal resolutions, of hourly $R=0.45$, daily $R=0.36$, and running-pentad $R=0.38$ (trend mode in Fig. 12b). The model performance with smoothed trend was acceptable for the same seven regions as Trend+ (6, 5, 2, 1 / 4, 3, 8). In Region 7, model $R$ values were also high but their range fell short of the rain-gauge values for hourly and daily extremes (first and second bars of trend mode in Fig. 12b). Model performance with NPGO index was acceptable for seven region (2, 1 / 7, 4, 3, 8). For TPI, this was only three regions (6, 5, 2). For SOI, this was five regions (6, 5, 2 / 3, 8), the same regions as ENSO+ excluding region 1. It seems that d4PDF produced appropriate relationships between hourly rainfall extremes and the slow components of Trend+, except for over Hokkaido. It should be used with caution when studying the ENSO and PDV due to the above-described biases.

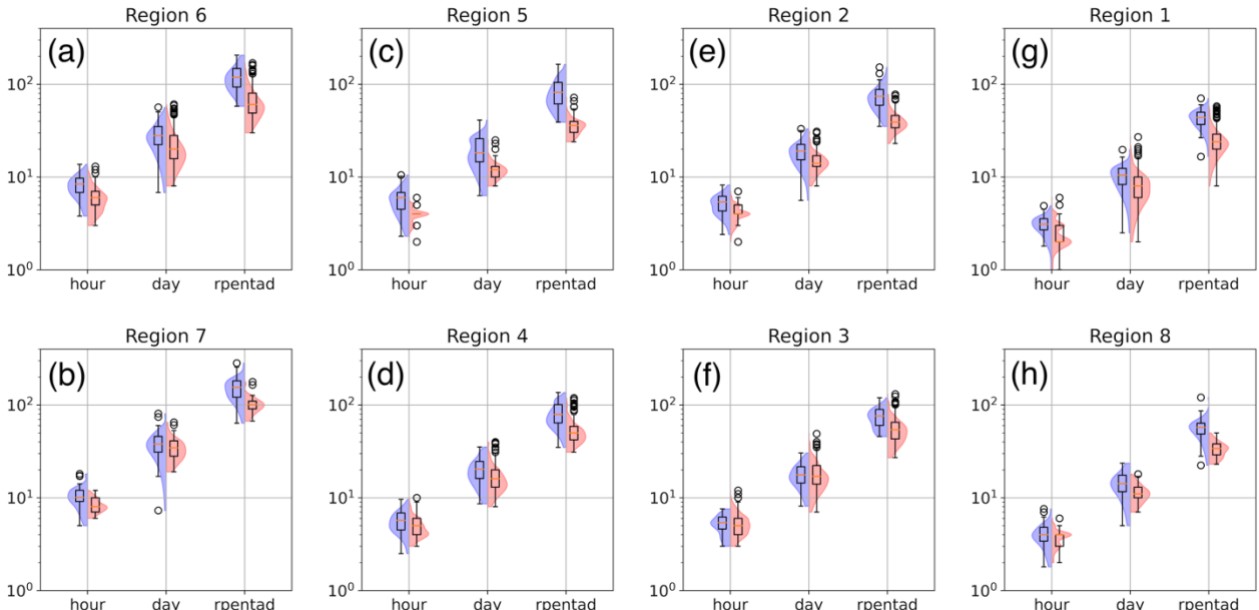

**Figure 6. The distribution of June–July 99th percentile hourly rainfall ("hour"), 90th percentile daily rainfall ("day"), and 90th percentile running-pentad rainfall ("rpentad") in mm. The eight analysis regions are arranged by their geographical location from southwest to northeast in Fig. 3f. Left side (blue shading) shows the distribution from rain-gauges, where each sample is the annual percentile value of the cluster. Right side (red shading) shows the distribution from d4PDF, where each sample is the annual percentile of each cluster co-located with the rain-gauges. Bar and whisker plots show the quantile values and circle markers showing outliers.**

575

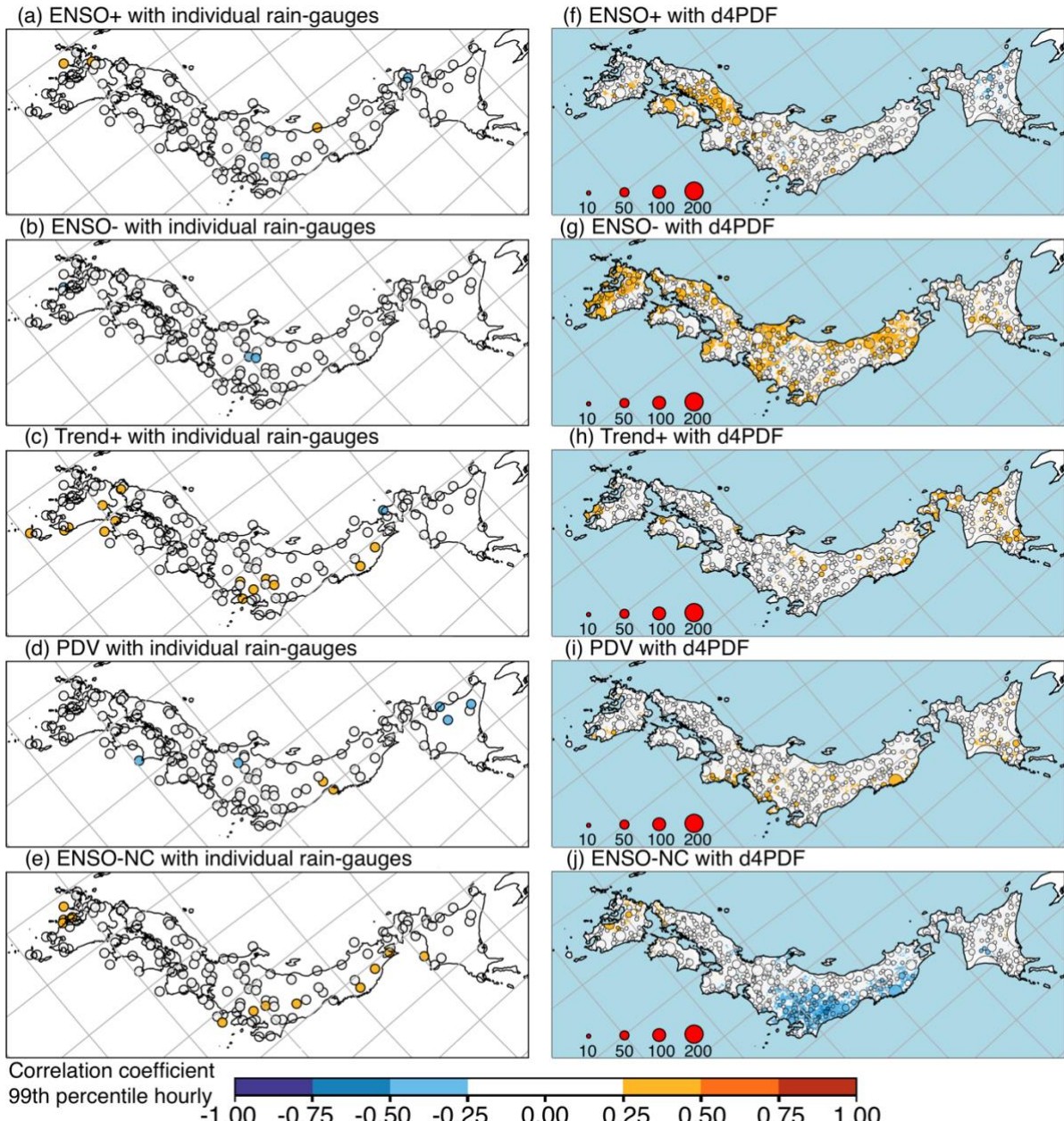

**Figure 7. Spearman correlation coefficients *R*'s for the 1952–2010 period between the scores of five Pacific SST modes (rows) and 99th percentile hourly rainfall from rain-gauges (left column) or d4PDF (right column). June–July rainfall extremes of individual rain-gauges with (a) ENSO+, (b) ENSO-, (c) Trend+, (d) PDV, (e) ENSO-NC. June–July rainfall extremes of d4PDF sets with (f) ENSO+, (g) ENSO-, (h) Trend+, (i) PDV, (j) ENSO-NC. In the right column, circle marker sizes reflect cluster sizes, while single points are shown as coloured dots. A blue mask was used to cover d4PDF data inside the ocean. Longitude-latitude gridlines are shown as grey lines; the map is a rotated projection.**

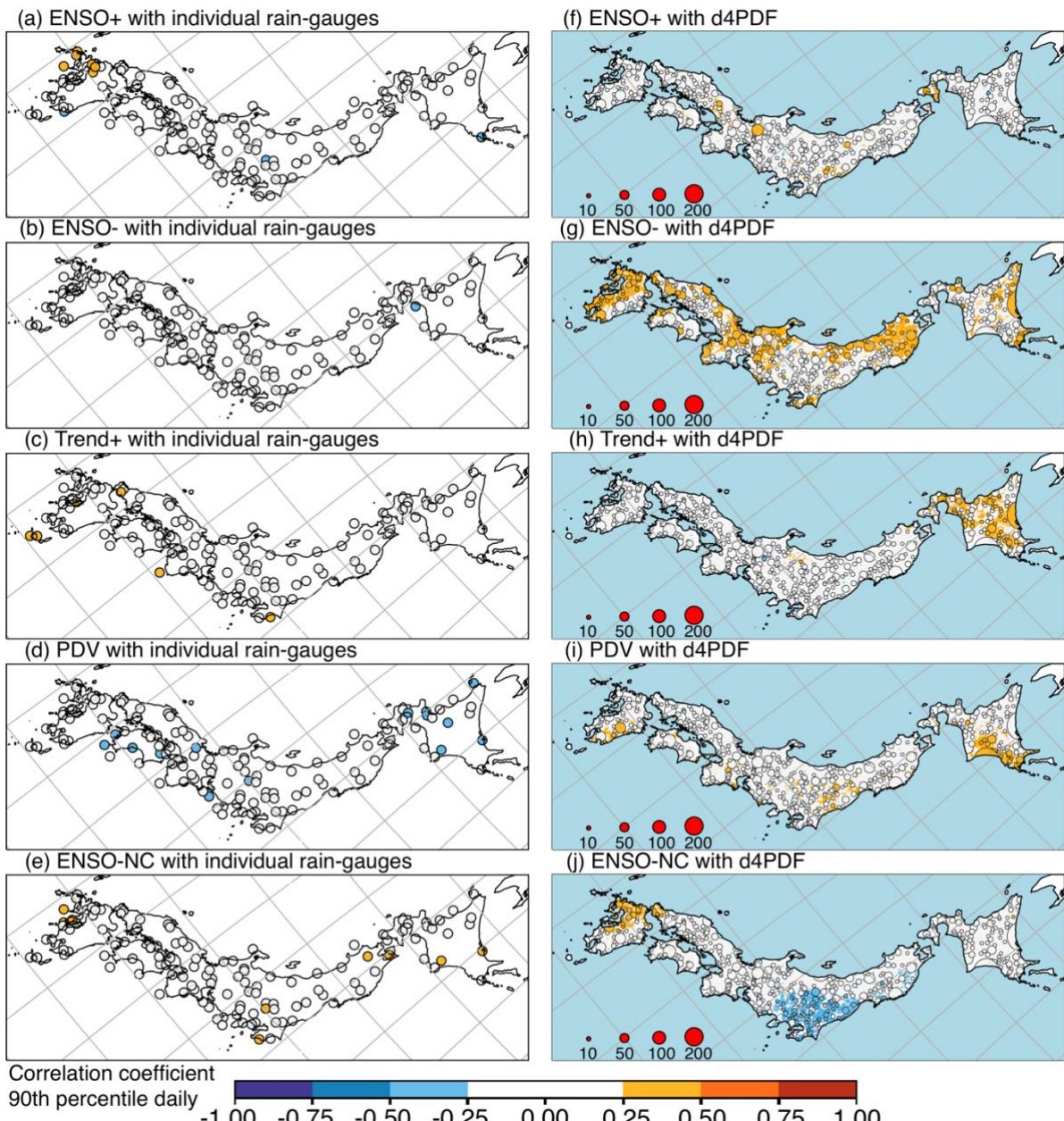

**Figure 8. Like as Fig. 7, except for seasonal 90th percentile daily rainfall.**

590

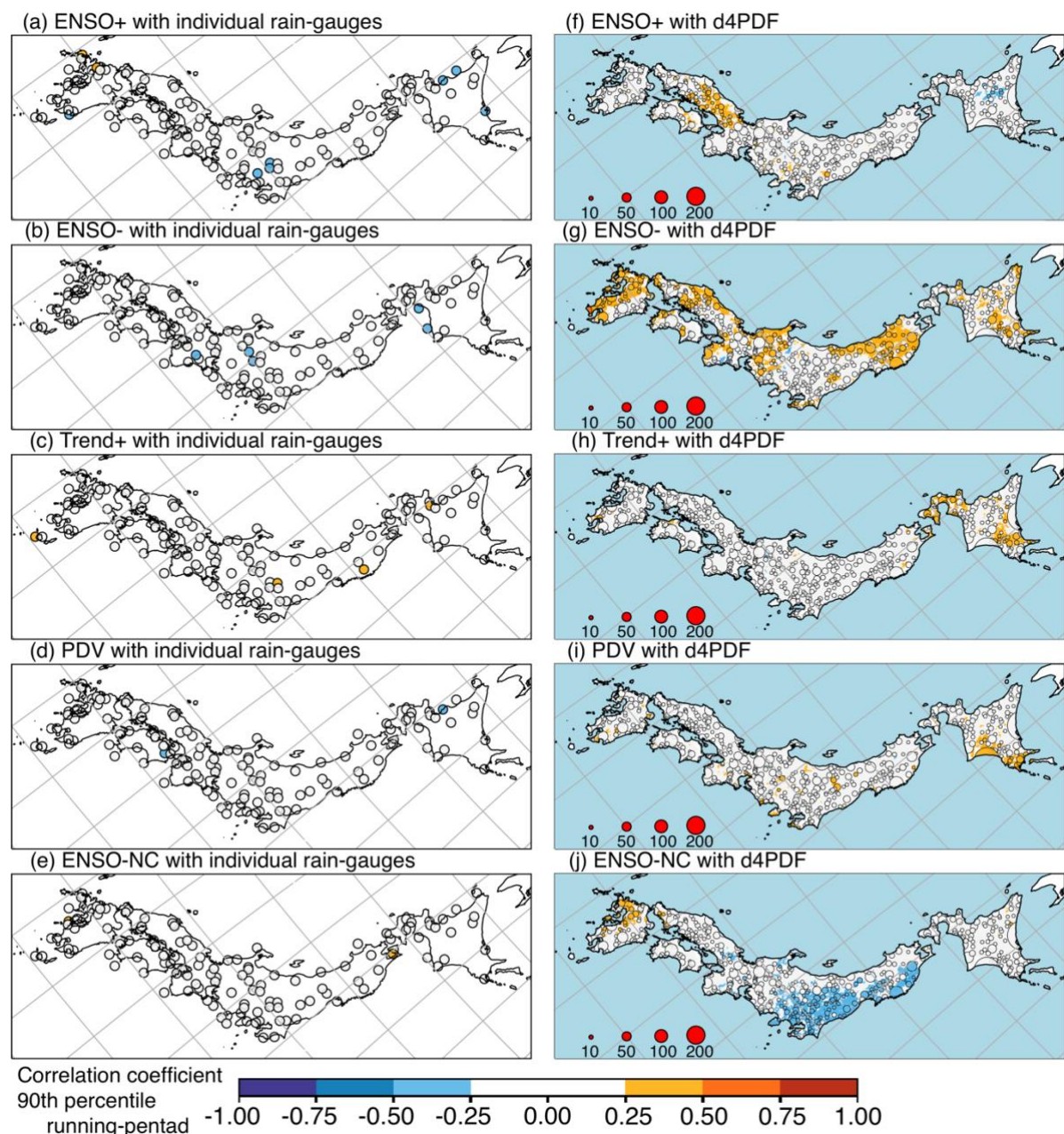

**Figure 9. Like Fig. 7, except for seasonal 90th percentile running-pentad rainfall.**

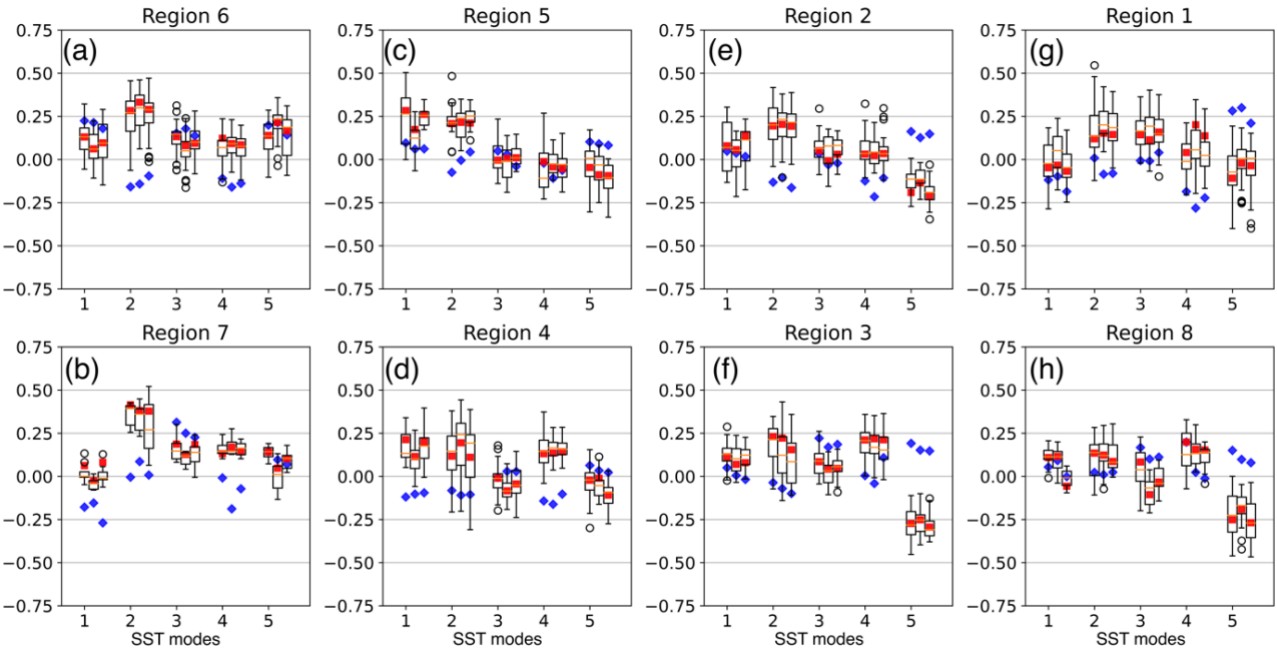

**Figure 10. Spearman correlation coefficients $R$'s between June–July rainfall extremes and the scores of the first five Pacific SST modes. The eight analysis regions are arranged by their geographical location from southwest to northeast in Fig. 3f. Three columns for each mode show the results for 99th percentile hourly, 90th percentile daily, and 90th percentile running-pentad rainfall. Blue diamonds represent the R's from rain-gauge cluster. Bar and whisker plots show the quantile values of the d4PDF clusters co-located with the rain-gauges, with circle markers showing outliers. Red boxes show the mean of d4PDF $R$'s weighted by cluster size.**

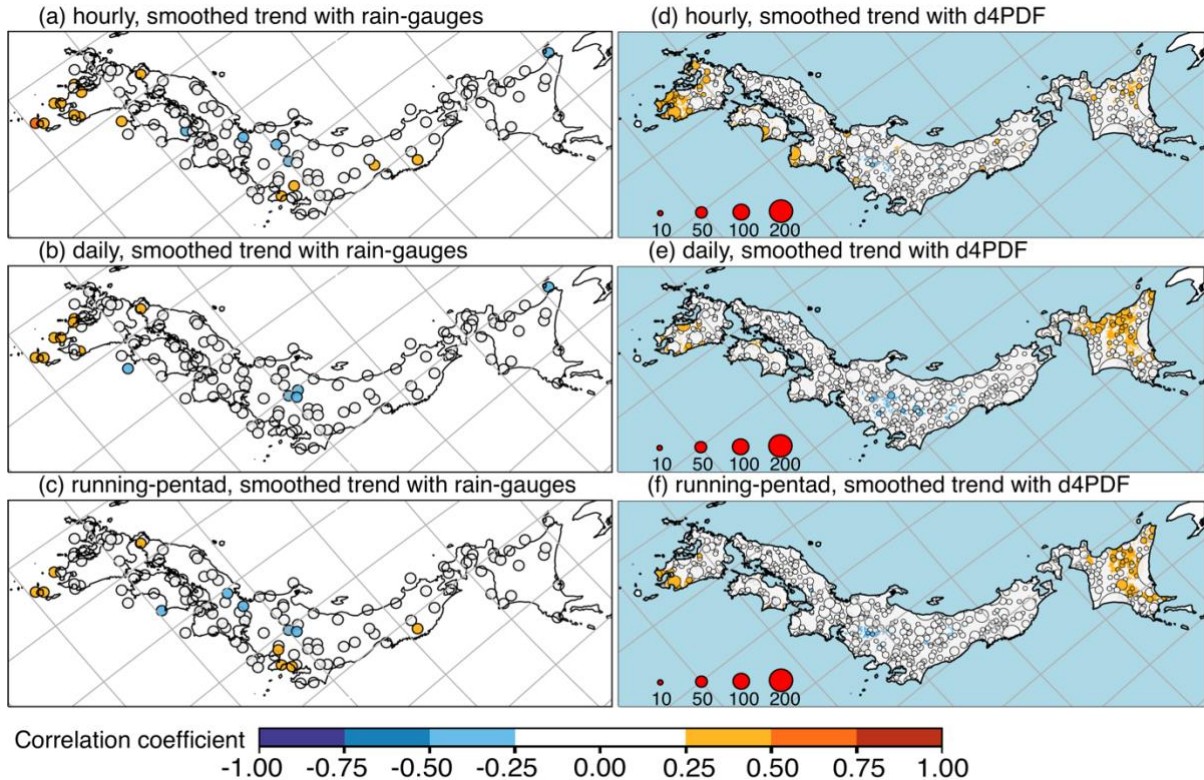

**Figure 11. Spearman correlation coefficients *R*'s for the 1957–2006 period between 11-year running mean of the Trend+ score June–July and rainfall extremes of different temporal resolution (rows), from individual rain-gauges (left column) or d4PDF (right column). From rain-gauges, (a) 99th percentile hourly rainfall, (b) 90th percentile daily rainfall, (c) 99th percentile running-pentad rainfall. From d4PDF, (d) 99th percentile hourly rainfall, (e) 90th percentile daily rainfall, (f) 99th percentile running-pentad rainfall. In the right column, circle marker sizes reflect cluster sizes, while single points are shown as coloured dots. A blue mask was used to cover d4PDF data inside the ocean. Longitude-latitude gridlines are shown as grey lines; the map is a rotated projection.**

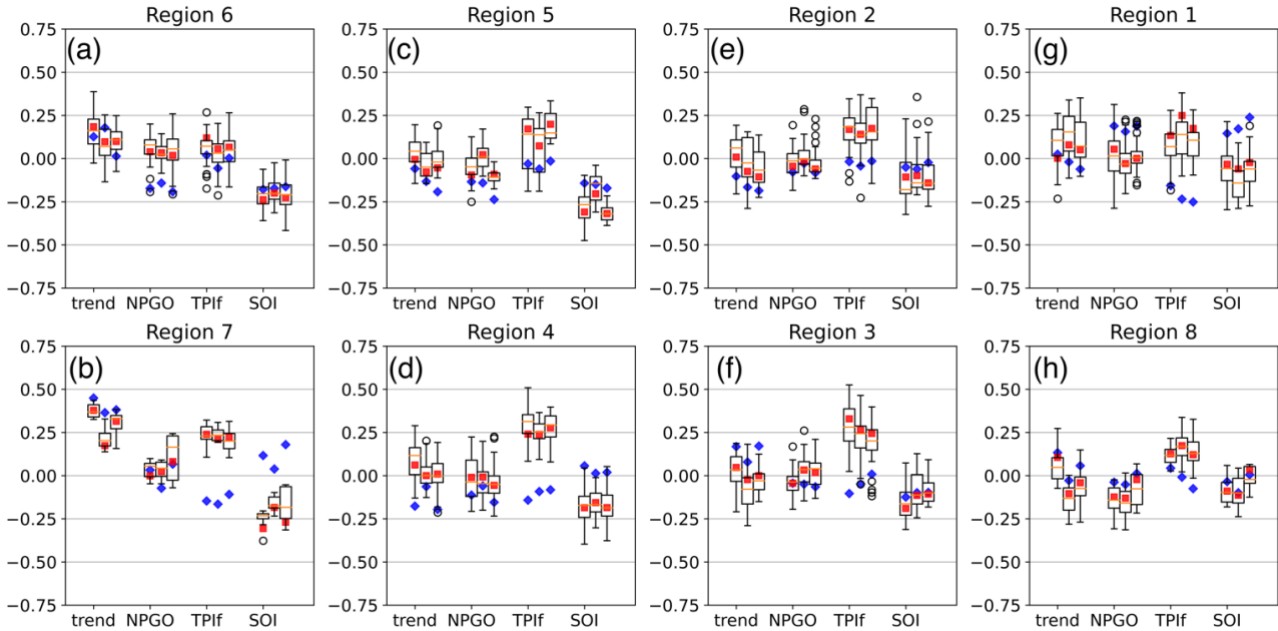

**Figure 12. Like Fig. 10 but with the 11-year running mean Trend+ score, NPGO index, TPI index (filtered version), and the SOI.**

## 4 Discussion

The lack of strong $R$'s between rainfall extremes and the Pacific SST modes indicates that the latter are not major controlling factors for extreme rainfall over Japan. These results are sensible in that extreme rainfall typically occurs in mesoscale systems which may be associated with certain synoptic scale conditions, but are much less likely to be directly influenced by hemispheric or planetary scale climate variability. In this section, rainfall extremes were correlated against indices reflecting the regional state of the monsoon near Japan, to check if there were stronger relationships with regional conditions. Rain-gauge rainfall extremes were correlated with monsoon indices calculated from JRA55, and these relationships were considered as "observations", shown on the left columns of Figs 13–18. d4PDF rainfall extremes were correlated with monsoon indices calculated from d4PDF-AGCM, shown on the right columns of Figs 13–18. Figure 19 summarises the results by analysis region.

Based on observations, rainfall extremes showed up to moderate correlation only with the seasonal anomaly of monsoon jet zonal strength ($\mu_{QU}$; Figs 14a, 16a, and 18a). Considering individual rain-gauges, domain-wide field significance was found for correlation with $\mu_{QU}$ using all three resolutions of hourly extremes (25 rain-gauges; Fig. 14a), daily extremes (36 rain-gauges; Fig. 16a), and running-pentad extremes (24 rain-gauges; a). The correlated rain-gauges were also concentrated along the Pacific coast of western Japan. This was reflected in significant cluster correlation in region 7 for all time resolutions with hourly $R=0.55$, daily $R=0.60$, and running-pentad $R=0.56$ (JQU in Fig. 19b); in region 3 for hourly and daily extremes with $R=0.31$ and $R=0.43$ respectively (first and second bars of JQU in Fig. 19f); region 4 for daily extremes with $R=0.28$ (second bar of JQU in Fig. 19d); region 6 for daily extremes with $R=0.29$ (second bar of JQU in Fig. 19a). This meant that cluster-wise field significance was found for daily correlation with $\mu_{QU}$.

The rest of the indices showed at most weak relationships, similar to the relationships with SST modes. However, rainfall extremes had better relationships with some monsoon indices in the sense that more rain-gauges showed significant relationships. Field significance was also found for correlation with the seasonal meridional anomaly of monsoon jet meridional strength ($\mu_{QV}$) using all three resolutions of hourly extremes (12 rain-gauges; Fig. 14b), daily extremes (15 rain-gauges; Fig. 16b), and running-pentad extremes (14 rain-gauges; b). The $\mu_{QV}$-correlated rain-gauges were concentrated along the Sea of Japan coast. Only region 1 showed significant correlation, significant at all three resolutions with hourly $R=0.28$, daily $R=0.36$, and running pentad $R=0.35$ (JQV in Fig. 19g).

Both $\sigma^2_{JLat}$ and $\sigma^2_{Q(U,V)}$ were indicators of monsoon jet instability, but $\sigma^2_{JLat}$ better related to rainfall extremes than $\sigma^2_{Q(U,V)}$. Field significance was found for anti-correlation with the seasonal variance of monsoon jet latitude ($\sigma^2_{JLat}$) using daily extremes (28 rain-gauges; Fig. 15d) and running-pentad extremes (15 rain-gauges; Fig. 17d), but not hourly extremes (11 rain-gauges only; Fig. 13d). The anti-correlated rain-gauges were concentrated along the Pacific coast of western Japan. The spatially widespread relationship was reflected in significant cluster anti-correlation in region 7 for daily and running-pentad extremes with $R=-0.39$ and $R=-0.33$ respectively (southern Kyushu; second and third bars of JLe in Fig. 19b); in region 4 for daily and running-pentad extremes with $R=-0.30$ and $R=-0.29$ respectively (Kansai; second and third bars of JLe in Fig. 16d); in region 3 for daily extremes only with $R=-0.31$ (Kanto; second bar of JLe in Fig. 19f). This meant that cluster-wise field significance was found for daily anti-correlation with $\sigma^2_{JLat}$.

Field significance was not found for the seasonal variance of monsoon jet strength ($\sigma^2_{Q(U,V)}$) despite the presence of a number of weakly anti-correlated rain-gauges for hourly extremes (Fig. 14d), daily extremes (Fig. 16d), and running-pentad extremes (d), because the number of significantly anti-correlated rain-gauges were only 11, 11 and 10, respectively. The anti-correlated rain-gauges were concentrated in southern Kyushu, and only region 7 showed significant anti-correlation, significant for all three resolutions with hourly $R=-0.36$, daily $R=-0.42$, and running-pentad $R=-0.42$ (JQe in Fig. 19b).

The relationships seen in d4PDF were much stronger than observed relationships. Only three rain-gauges in southwest Kyushu showed moderate relationships and only with $\mu_{QU}$, two for hourly and daily extremes (Figs 14a and 16a), and a different one for running-pentad extremes (Figs 14a and 18a). d4PDF showed up to moderate relationships between rainfall extremes and five monsoon indices $\mu_{QV}$, $\sigma^2_{JLat}$, $\mu_{QU}$, $\mu_{JLat}$, and $\sigma^2_{Q(U,V)}$, in order of influence. The spatial patterns for $\mu_{QV}$ and

$\mu_{QU}$ were similar to each other, and between temporal resolutions (Figs 14d–e, 16d–e, 18d–e). This was correlation along the Sea of Japan coast, correlation along the Pacific coast west of topography, but anti-correlation along the Pacific coast east of topography. The spatial patterns for $\sigma^2_{JLat}$ was similar between temporal resolutions (Figs 13h, 15h, 17h). This was the reversed pattern with anti-correlation along the Sea of Japan coast, anti-correlation along the Pacific coast west of topography, but correlation along the Pacific coast east of topography. The most influential three indices $\mu_{QV}$, $\sigma^2_{JLat}$, and $\mu_{QU}$ for d4PDF were also the three indices that showed field-significant relationships for observation. However, the observed order of importance was $\mu_{QU}$, $\sigma^2_{JLat}$, and $\mu_{QV}$.

The other two monsoon indices showing moderate correlations with rainfall extremes had better relationships with hourly and pentad extremes than with daily extremes. For $\mu_{JLat}$, areas with moderate anti-correlation were not spatially widespread, and occurred only with scattered clusters in southern Kyushu or Chugoku for hourly and running-pentad extremes (Figs 13g and 17g, but not Fig. 15g). Weak anti-correlation was spatially widespread over western Japan so this was a relatively influential mode for d4PDF, but not for observation. For $\sigma^2_{Q(U,V)}$, moderate correlation was concentrated over eastern Kyushu only for hourly extremes (Fig. 14f). Weak correlation was seen over eastern Kyushu, Chugoku, some parts along the Sea of Japan coast for hourly and pentad extremes, but only over eastern Kyushu for daily extremes.

The remaining two monsoon indices related to the monsoon front latitude $\sigma^2_{FLat}$ and $\mu_{FLat}$ showed weak relationships that were spatially more widespread than observation. For $\sigma^2_{FLat}$, this was clearly so with weak anti-correlation over western Kyushu and weak correlation along the Pacific coast of eastern Japan for all three temporal resolutions (Figs 13f, 15f, 17f). For $\mu_{FLat}$, this was less clearly so and only over eastern Japan. Clusters with weak anti-correlation were concentrated over the Kanto region, and most widespread for hourly extremes (Fig. 13f, vs Figs 15f and 17f). Weak correlation was seen along the Sea of Japan coast, but only widespread for daily extremes (Fig. 15f, vs Figs 13f and 17f).

In both observations and d4PDF, rainfall extremes had stronger or more spatially widespread relationships with some of the regional monsoon indices than with Pacific SST modes. This supported the hypothesis that Pacific SST modes act indirectly through the regional monsoon. Hence, correlations between the monsoon indices and standardised principal components of the Pacific SST modes were calculated (Table 2). Relationships from both JRA55 and d4PDF were at most weak, and we propose that multiple aspects of influence through the monsoon indices need to stack for a relationship to be detected between the SST modes and extreme rainfall. This method of explanation will first be applied to the ENSO-related SST modes, to explain their relationships with rainfall extremes in d4PDF were excessive compared to rain-gauges. Then the method will be less successfully applied to PDV and Trend+, to explain why relationships with rainfall extremes were seen in rain-gauges, but not in d4PDF.

ENSO+ was significantly correlated with $\sigma^2_{Q(U,V)}$ for both d4PDF ($R=0.31$) and JRA55 ($R=0.30$). There was significant anti-correlation only for d4PDF, with $\mu_{QV}$ ($R=-0.28$), $\mu_{JLat}$ ($R=-0.29$), and $\mu_{FLat}$ (R=-0.34), but not for JRA55 ($R=0.04$, $R=-0.20$, and $R=-0.15$, respectively). The $\mu_{JLat}$ $R$ values were similar between d4PDF and JRA55, but only d4PDF showed a spatially widespread relationship between $\mu_{JLat}$ and extreme rainfall (panels c vs d for Figs 13, 15, and 17), so $\mu_{JLat}$ was influential only for d4PDF. d4PDF displayed the stacked spatiotemporal signals of $\sigma^2_{Q(U,V)}$ and reversed $\mu_{JLat}$ in the ENSO+ signals. Firstly, ENSO+ also showed weak correlation over eastern Kyushu only for hourly extremes (Fig. 7f) but not otherwise (Figs 8f and 9f); $\sigma^2_{Q(U,V)}$ showed up to moderate correlation over eastern Kyushu only for hourly extremes (Fig. 14f) but weak correlation otherwise (Figs 16f and 18f); reversed $\mu_{JLat}$ showed up to moderate correlation over Kyushu only for hourly extremes (reversed Fig. 13g) but weak correlation otherwise (reversed Figs 15g and 17g). Secondly, ENSO+ also showed weak correlation only for hourly and running-pentad extremes (Figs 7f and 9f) but not for daily extremes (Fig. 8f); $\sigma^2_{Q(U,V)}$ showed weak correlation over Chugoku only for hourly and running-pentad extremes (Figs 14f and 18f) but not daily extremes (Fig. 16f); reversed $\mu_{JLat}$ showed up to moderate correlation over Chugoku only for hourly and running-pentad extremes (reversed Figs 13g and 17g) but weak correlation for daily extremes (reversed Fig. 15g). The signals of reversed $\mu_{QV}$, reversed $\mu_{FLat}$, and western Kyushu part of reversed $\mu_{JLat}$ seemed to have "cancelled out". Reversed $\mu_{QV}$ showed anti-correlation over western Kyushu and along the Sea of Japan coast (Figs. 14e, 16e, 18e); reversed $\mu_{FLat}$ showed correlation along the Sea of Japan coast (Figs 13e, 15e, 17e); reversed $\mu_{JLat}$ showed correlation over western Kyushu (Figs 13g, 15g, 17g). Reversed $\mu_{QV}$ showed correlation over Kanto; reversed $\mu_{FLat}$ showed anti-correlation over Kanto. Unlike the stacking of

signals with the same direction, it was more difficult to decide whether antagonising signals would "cancel". Two factors
were involved: the strength of the relationship between SST modes and the regional monsoon, as well as the strength of the
relationship between the regional monsoon modes and extreme rainfall.

ENSO- was significantly correlated with $\mu_{QU}$ for both d4PDF ($R$=0.46) and JRA55 ($R$=0.30). There was significant
correlation with $\mu_{QV}$ for d4PDF only ($R$=0.32), but not for JRA55 ($R$=0.02). There was significant anti-correlation with
$\sigma^2_{Q(U,V)}$ for JRA55 only ($R$=-0.30), but not d4PDF ($R$=-0.11). There was significant anti-correlation for d4PDF with $\sigma^2_{JLat}$
($R$=-0.31), $\mu_{JLat}$ ($R$=-0.32), and $\mu_{FLat}$ ($R$=-0.30), but not for JRA55 ($R$=-0.05, $R$=0.07, and $R$=-0.20, respectively). For
observation, the stacking of correlation over southern Kyushu from $\mu_{QU}$ (Figs 14a, 16a, 18a) and reversed $\sigma^2_{Q(U,V)}$ (Figs 14c,
16c, 18c) suggested that correlated rain-gauges should be seen over southern Kyushu, but this was not the case (Figs 7b, 8b,
and 9b). However, stacking could explain d4PDF's excessive correlation between ENSO- and rainfall extremes. The three
patterns of $\mu_{QU}$ (Figs 14d, 16d, 18d), $\mu_{QV}$ (Figs 14e, 16e, 18e), and reversed $\sigma^2_{JLat}$ (reversed Figs 13h, 15h, 17h) were similar,
with correlation along the Sea of Japan coast, correlation along the Pacific coast west of topography, but anti-correlation
along the Pacific coast east of topography. Over western Japan, the anti-correlation was antagonised by the reversed $\mu_{JLat}$
signal of correlation (reversed Figs 13g, 15g, 17g). Over eastern Japan particularly Kanto, the anti-correlation was
antagonised by the reversed $\mu_{FLat}$ signal of correlation (reversed Figs 13e, 15e, 17e). This produced the d4PDF spatial pattern
of correlation along the Sea of Japan coast and along the Pacific coast west of topography, but no relationship along the
Pacific coast east of topography (Figs 7g, 8g, 9g). A few moderately anti-correlated clusters were seen over Kanto for hourly
extremes (Fig. 7g)

ENSO-NC showed only significant relationships with d4PDF. This was correlation with $\mu_{QU}$ ($R$=0.31, vs JRA55 $R$=0.08),
anti-correlation with $\sigma^2_{JLat}$ ($R$=-0.36, vs JRA55 $R$=-0.01), and anti-correlation with $\sigma^2_{FLat}$ ($R$=-0.27, vs JRA55 $R$=0.19). Since
$\sigma^2_{FLat}$ was not influential for JRA55, the higher magnitude of $R$ for JRA55 did not matter. The two patterns of $\mu_{QU}$ (Figs 14d,
16d, 18d) and reversed $\sigma^2_{JLat}$ (reversed Figs 13h, 15h, 17h) were similar, as earlier described. This combined pattern stacked
further with reversed $\sigma^2_{FLat}$, which showed correlation over western Kyushu and anti-correlation along the Pacific coast of
eastern Japan (reversed Figs 13f, 15f, 17f). This produced the d4PDF spatial pattern of correlation over western Kyushu, and
anti-correlation over eastern Japan (Figs 7j, 8j, 9j).

PDV showed no significant relationships with monsoon indices for either JRA55 or d4PDF. For JRA55, there were still two
$R$'s of stronger (but not significant) magnitude, with $\mu_{QV}$ ($R$=-0.24, vs d4PDF $R$=0.09) and $\sigma^2_{JLat}$ ($R$=0.20, vs d4PDF $R$=0.06).
For d4PDF, there was only one $R$ of stronger magnitude, with $\mu_{QU}$ ($R$=-0.20, vs JRA55 $R$=-0.12). Stacking could still explain
the observed relationship between PDV and rainfall extremes, based on the spatiotemporal signals of $\sigma^2_{JLat}$. Rain-gauges
showed field significant anti-correlation with PDV only for daily extremes only (13 rain-gauges; Fig. 8d) but not otherwise
(Figs 7d, 9d); with $\sigma^2_{JLat}$ cluster-wise and gauge-wise field significant anti-correlation for daily extremes (28 rain-gauges and
three clusters; Fig 15d), gauge-wise field significant anti-correlation for running-pentad extremes (15 rain-gauges but only
one cluster; Fig 17d), and no field significance for hourly extremes (only 11 rain-gauges and no clusters; Fig 13d). Rain-
gauges showing anti-correlation with PDV were located over western Japan and Hokkaido (Fig. 8d); rain-gauges showing
anti-correlation with $\sigma^2_{JLat}$ were located along the Pacific coast of western Japan (Fig 15d), stacking with reversed $\mu_{QV}$ with
anti-correlated rain-gauge along the Sea of Japan coast of western Japan and over Hokkaido (Fig. 16b).

Trend+ showed there only one statistically significant $R$ amongst the regional monsoon indices, with $\sigma^2_{FLat}$ for JRA55
($R$=0.29, vs d4PDF $R$=-0.08). However, $\sigma^2_{FLat}$ was not influential for JRA55. $R$'s of stronger magnitude were with $\mu_{QV}$ for
both JRA55 and d4DF (both $R$=0.21), with JRA55 only ($R$=0.20, vs d4PDF $R$=0.04), and with for d4PDF only ($R$=0.19, vs
JRA55 R=-0.10). For observation, although $\mu_{QU}$ and $\mu_{QV}$ patterns might stack in terms of domain-wide correlation, the
significantly correlated rain-gauges were not co-located, with rain-gauges for $\mu_{QU}$ and $\mu_{QV}$ concentrated along the Pacific
coast and Sea of Japan coast, respectively. Furthermore, correlation with Trend+ was field significant only for hourly
extremes (12 rain-gauges; Fig 6c), but correlation with $\mu_{QU}$ was best for daily extremes with cluster-wise and field
significance (36 rain-gauges and four clusters; Fig 16a). For d4PDF, correlation with Trend+ was concentrated over
Hokkaido, and did not resemble the spatial patterns of $\mu_{QV}$ or $\sigma^2_{Q(U,V)}$. Action through the regional monsoon front or regional
monsoon jet does not seem to be a good mechanism to explain the influence of Trend+ on hourly rainfall extremes.

Finally, an explanation is proposed for why d4PDF showed stronger and spatially more widespread relationships between rainfall extremes and most of the regional monsoon indices. Considering the three most influential regional monsoon indices of $\mu_{QU}$, $\mu_{QV}$ and $\sigma^2_{JLat}$, we observe that $\mu_{QU}$ and $\sigma^2_{JLat}$ had reversed relationships with rainfall extremes. However, observed relationships were uniform, while d4PDF relationships were bipolar and showed strong topographic influence. This suggested that one cause was the movement of the regional monsoon, with d4PDF having a more stable location that allowed topographic influences to strongly manifest.

Figure 20 shows the bi-weekly location of the regional monsoon for JRA55 and d4PDF. In early June, the monsoon was located at similar latitudes for both JRA55 and d4PDF, about to move over the study domain (Fig. 20a vs Fig. 20e). In Late June, the monsoon had moved over southern Kyushu for both JRA55 and d4PDF, but even at this time it was slightly more north where it passed over Kyushu for JRA55 (Fig. 20b vs Fig. 20f). In early July, the JRA55 monsoon had passed over Kyushu, and lay over the Sea of Japan coast of western Japan. Over eastern Japan, it lay south of the Pacific coast. In contrast, the d4PDF monsoon was still over northwest Kyushu and lay between Chugoku and Shikoku island in western Japan. Over eastern Japan, it lay along the Pacific Coast. In late July, the JRA55 monsoon had reached the Korean peninsula and lay north of western Japan, then diagonally southeast over eastern Japan into the Pacific. In contrast, the d4PDF monsoon still had the position similar to JRA55's early July position. It had a more zonal orientation extended further over eastern Japan, rather than off the coast as in JRA55. Hence, the monsoon moved two weeks more slowly across the three large Japanese islands for d4PDF.

All other things being equal, the same synoptic disturbances travelling from the west would have a larger influence on the study domain in d4PDF, simply from the land regions of Japan capturing more rainfall from them in d4PDF. At least some of the differences between observation and d4PDF originated from the global model driving the regional model. This did not exclude other differences that might have originated from the regional model, but improvement of the monsoon movement in the global model would probably also improve extreme rainfall's global and regional teleconnections in d4PDF. The difference in monsoon position was quite minor, considering that it was only a regional spatiotemporal bias of about 2 weeks and 2 degrees latitude in a global climate model. The difficulty to reproduce correct teleconnections is likely due to the unfortunate combination of a zonally-aligned physical phenomenon with a zonally-aligned geographically narrow island.

**Table 2. Spearman correlation coefficients $R$'s for the 1958–2010 period between the standardised principal components of five Pacific SST modes (rows) and the seven regional monsoon indices (columns), for JRA55 and d4PDF (sub-rows)**

| SST mode | Data | Monsoon Front Latitude | | Monsoon Jet Latitude | | Monsoon Jet Water Vapour Flux | | |
|---|---|---|---|---|---|---|---|---|
| | | | | | | anomaly | | |
| | | anomaly $\mu_{FLat}$ | variance $\sigma^2_{FLat}$ | anomaly $\mu_{JLat}$ | variance $\sigma^2_{JLat}$ | zonal $\mu_{QU}$ | meridional $\mu_{QV}$ | variance $\sigma^2_{Q(U,V)}$ |
| ENSO+ | JRA55 | -0.15 | -0.06 | -0.20 | 0.15 | -0.20 | 0.04 | **0.30** |
| | d4PDF | **-0.34** | 0.15 | **-0.29** | -0.06 | -0.07 | **-0.28** | **0.31** |
| ENSO- | JRA55 | -0.20 | -0.15 | 0.07 | -0.05 | **0.30** | 0.02 | **-0.30** |
| | d4PDF | **-0.30** | 0.04 | **-0.32** | **-0.32** | **0.46** | **0.32** | -0.11 |
| Trend+ | JRA55 | -0.08 | **0.29** | -0.16 | -0.14 | 0.20 | 0.21 | -0.10 |
| | d4PDF | -0.04 | -0.03 | -0.02 | 0.01 | 0.04 | 0.21 | 0.19 |
| PDV | JRA55 | 0.03 | 0.16 | 0.14 | 0.20 | -0.12 | -0.24 | -0.02 |
| | d4PDF | 0.05 | -0.03 | -0.03 | 0.06 | -0.20 | 0.09 | 0.10 |
| ENSO-NC | JRA55 | -0.10 | 0.19 | -0.07 | -0.01 | 0.08 | -0.08 | -0.12 |
| | d4PDF | -0.06 | **-0.27** | -0.03 | **-0.36** | **0.31** | 0.12 | -0.11 |

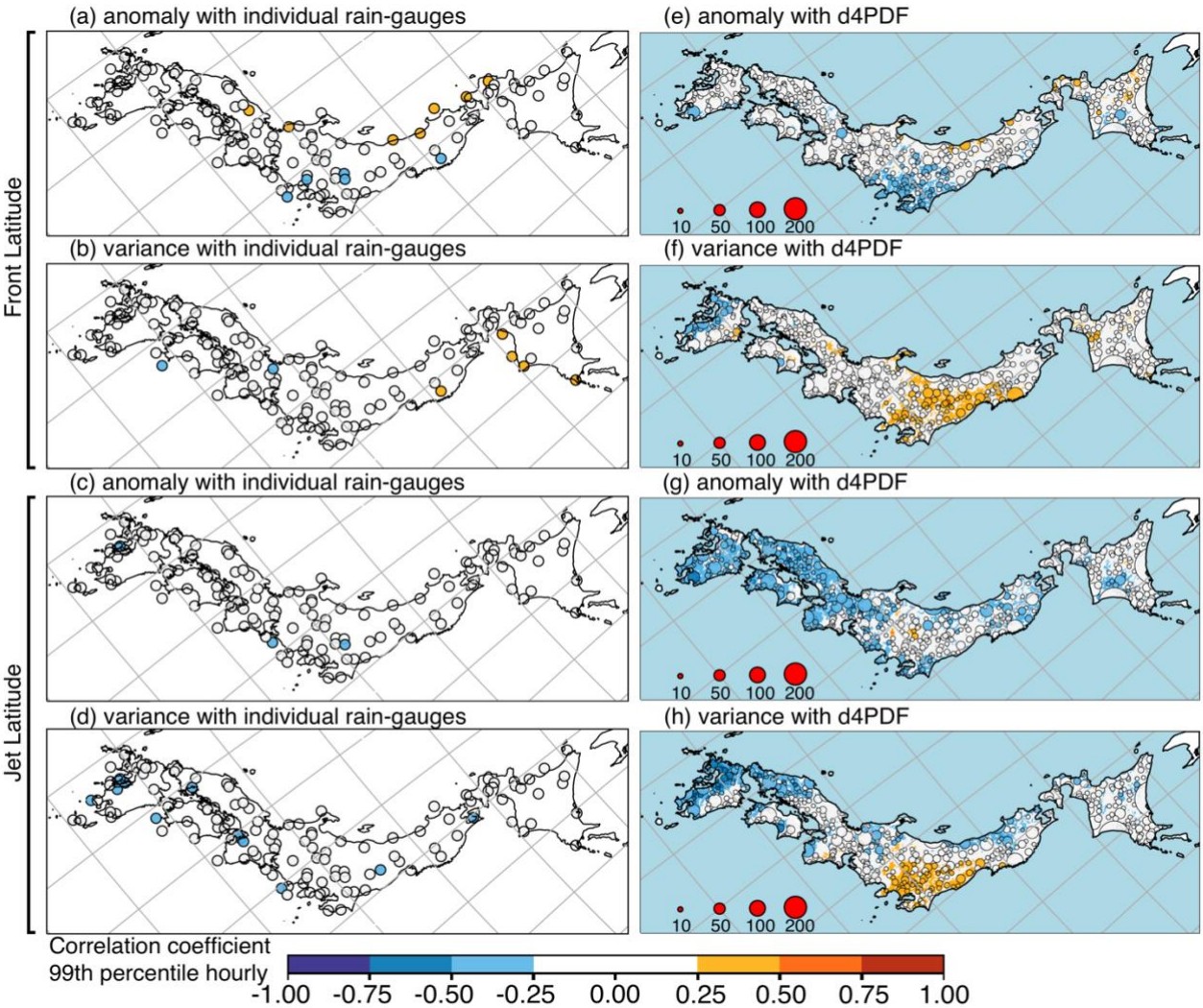

**Figure 13. Spearman correlation coefficients $R$'s for the 1958–2010 period between monsoon indices related to position (rows) and June–July 99th percentile hourly rainfall from rain-gauges (left column) or d4PDF (right column). Rainfall extremes of individual rain-gauges with JRA55 indices; (a) seasonal anomaly of monsoon front latitude $\mu_{FLat}$, (b) seasonal variance of monsoon front latitude $\sigma^2_{FLat}$, (c) seasonal anomaly of monsoon jet latitude $\mu_{JLat}$, (d) seasonal variance of monsoon jet latitude $\sigma^2_{JLat}$. Rainfall extremes of d4PDF sets with d4PDF indices; (e) $\mu_{FLat}$, (f) $\sigma^2_{FLat}$, (g) $\mu_{JLat}$, (h) $\sigma^2_{JLat}$. In the right column, circle marker sizes reflect cluster sizes, while single points are shown as coloured dots. A blue mask was used to cover d4PDF data inside the ocean. Longitude-latitude gridlines are shown as grey lines; the map is a rotated projection.**

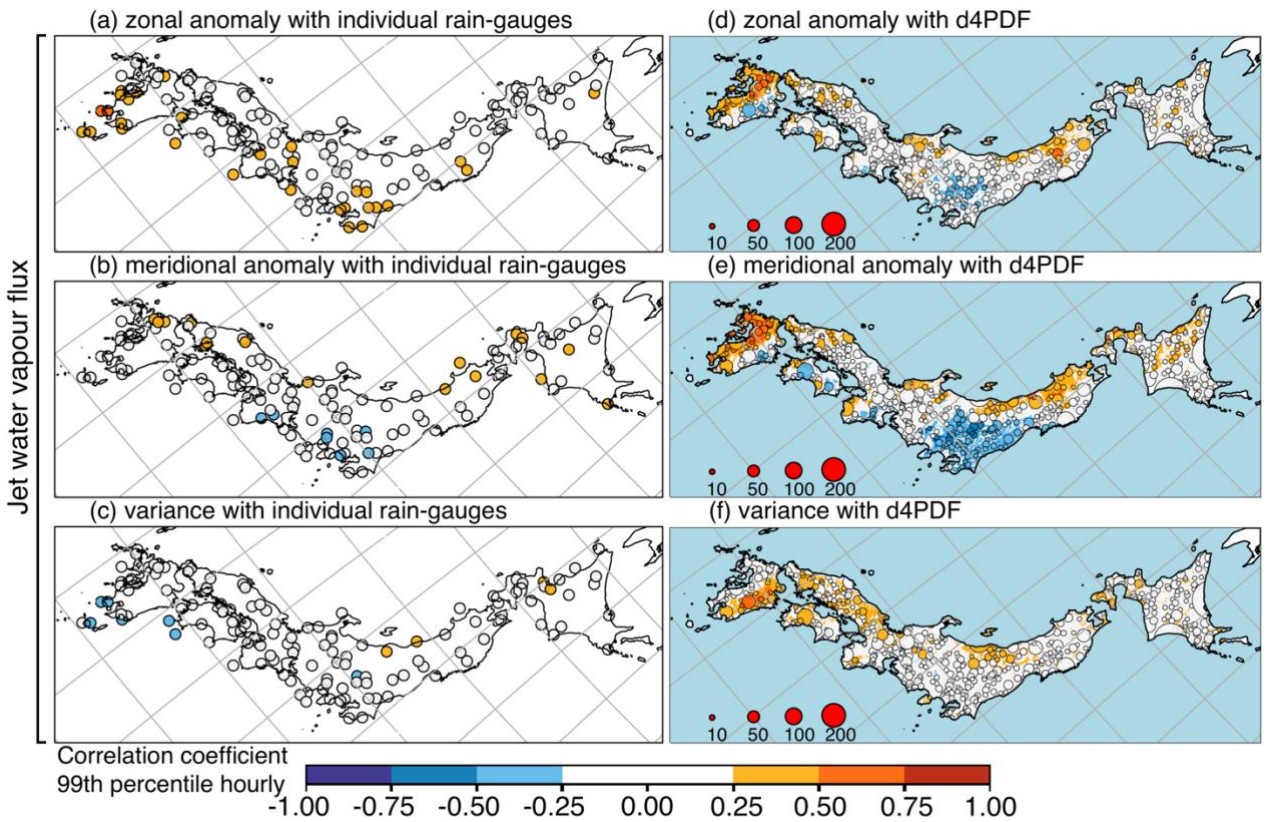

**Figure 14. Like Fig. 13, but with monsoon indices related to the strength of the monsoon jet. Rainfall extremes of individual rain-gauges with JRA55 indices; (a) seasonal anomaly of the monsoon jet zonal water vapour flux $\mu_{QU}$, (b) seasonal anomaly of the monsoon jet meridional water vapour flux $\mu_{QV}$, (c) seasonal variance of the monsoon jet water vapour flux $\sigma^2_{Q(U,V)}$. Rainfall extremes of d4PDF sets with d4PDF indices; (d) $\mu_{QU}$, (e) $\mu_{QV}$, (f) $\sigma^2_{Q(U,V)}$.**


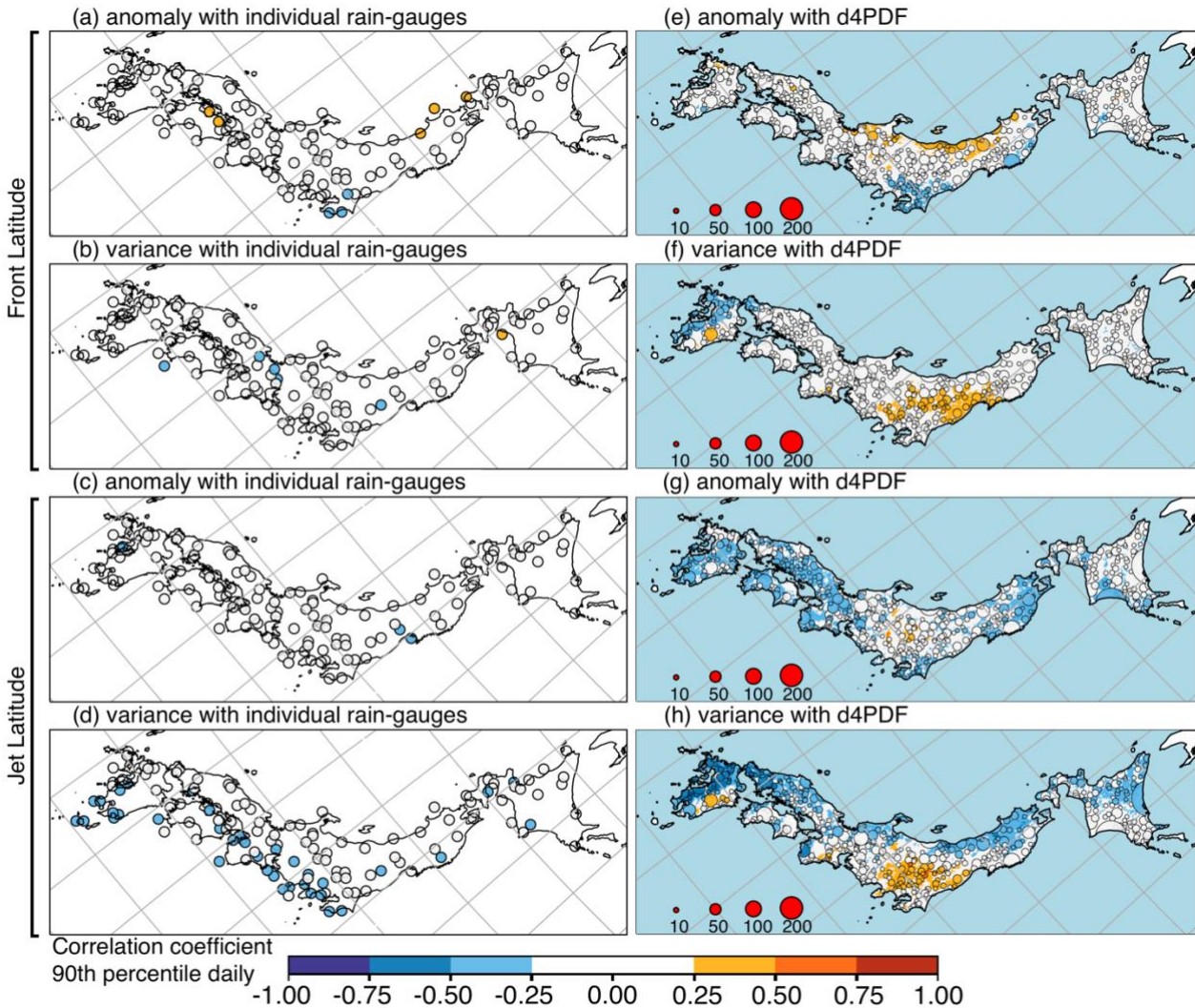

**Figure 15. Like Figure 13, but for 90th percentile daily rainfall.**

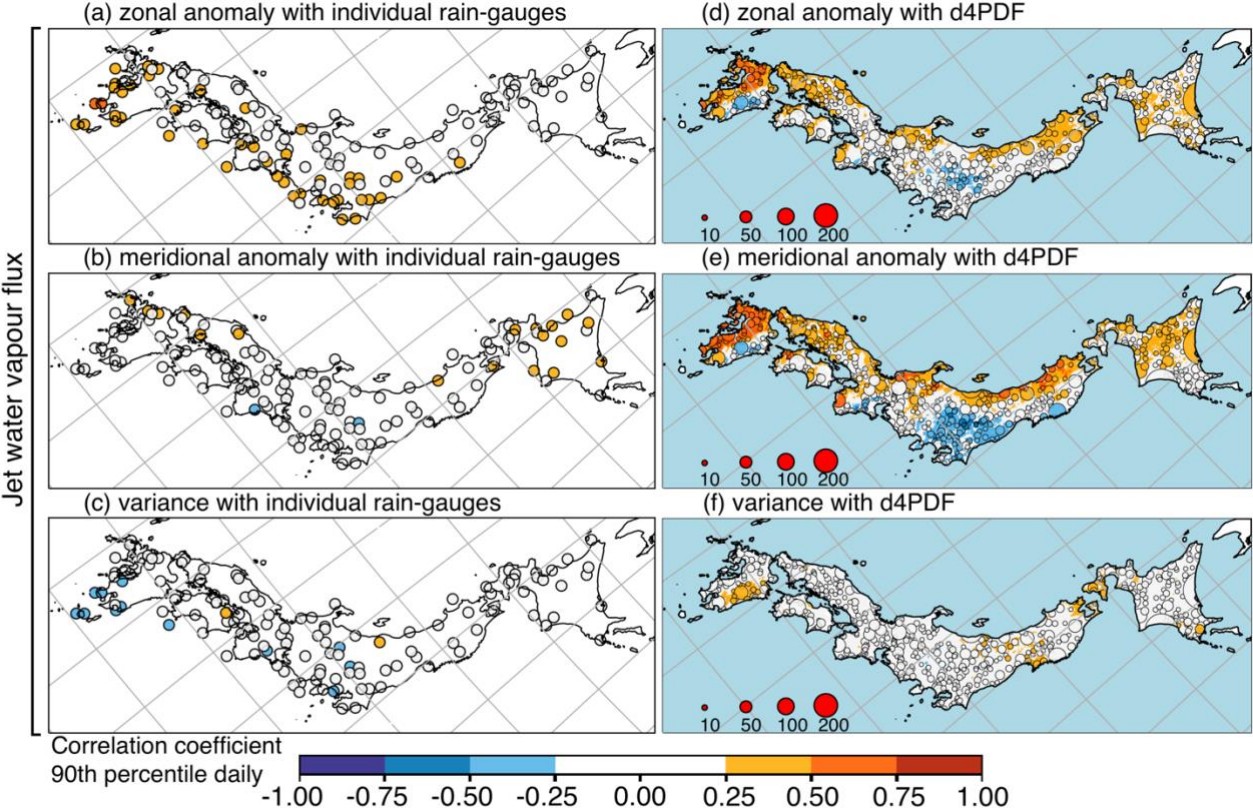

Figure 16. Like Figure 14, but for 90th percentile daily rainfall.

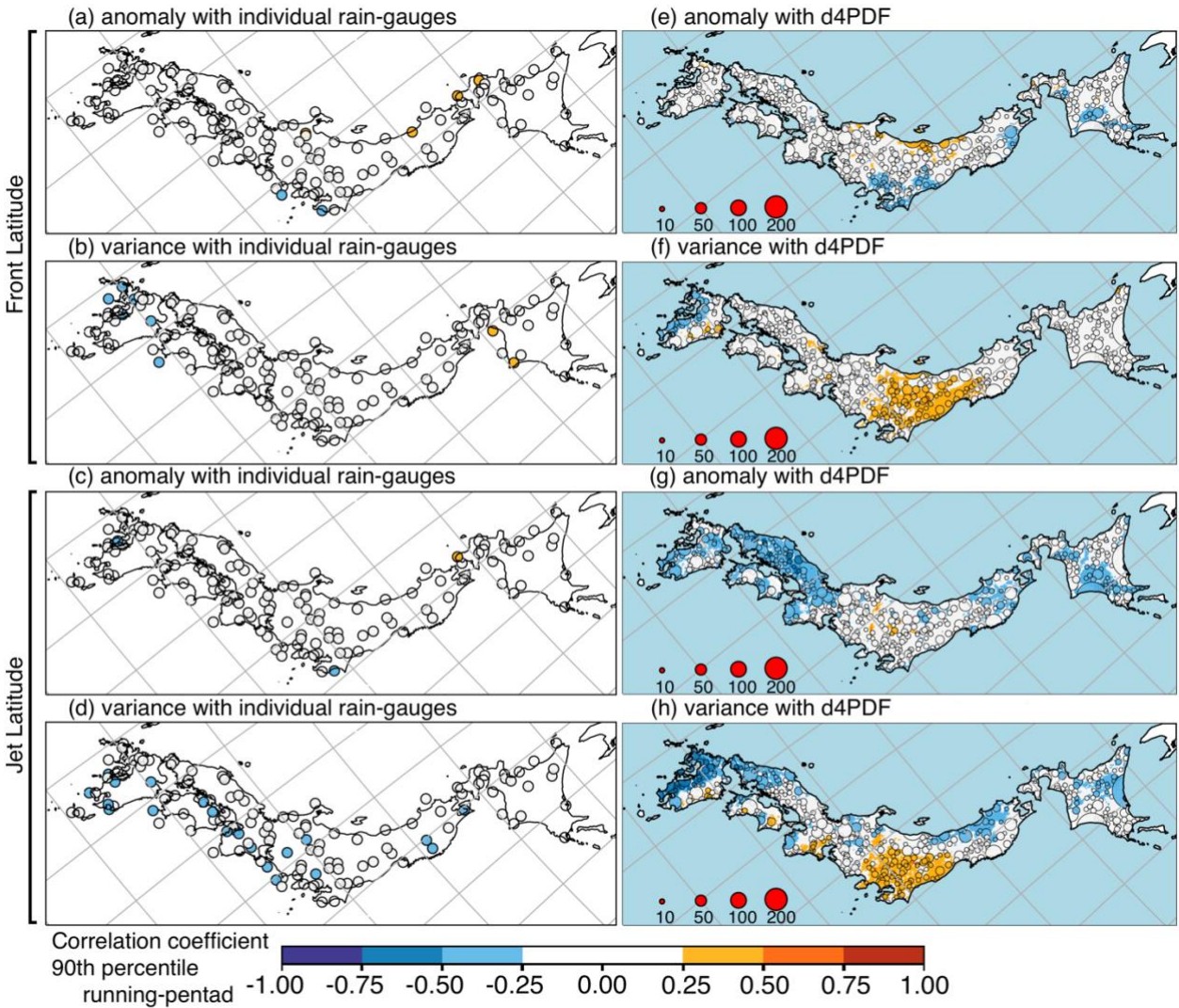

**Figure 17. Like Figure 13, but for 90th percentile running-pentad rainfall.**

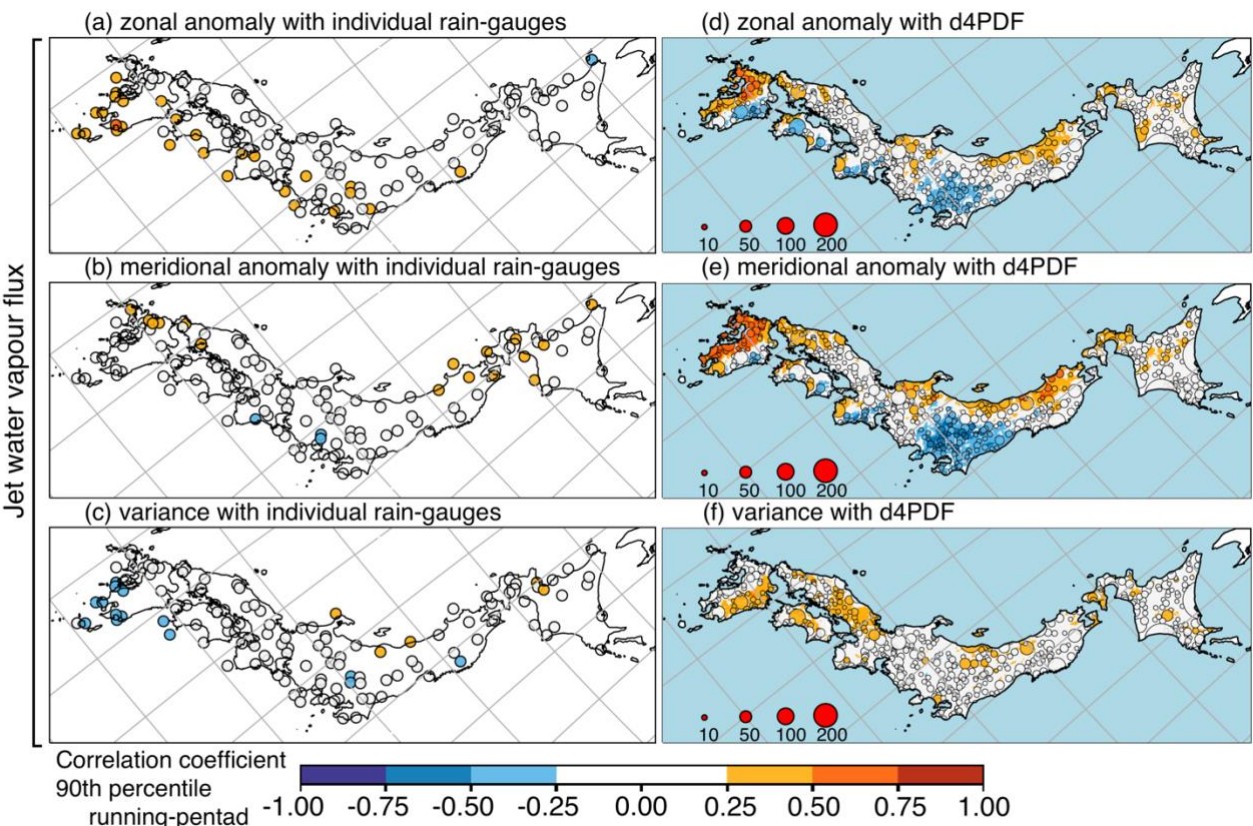

**Figure 18. Like Figure 14, but for 90th percentile running-pentad rainfall.**

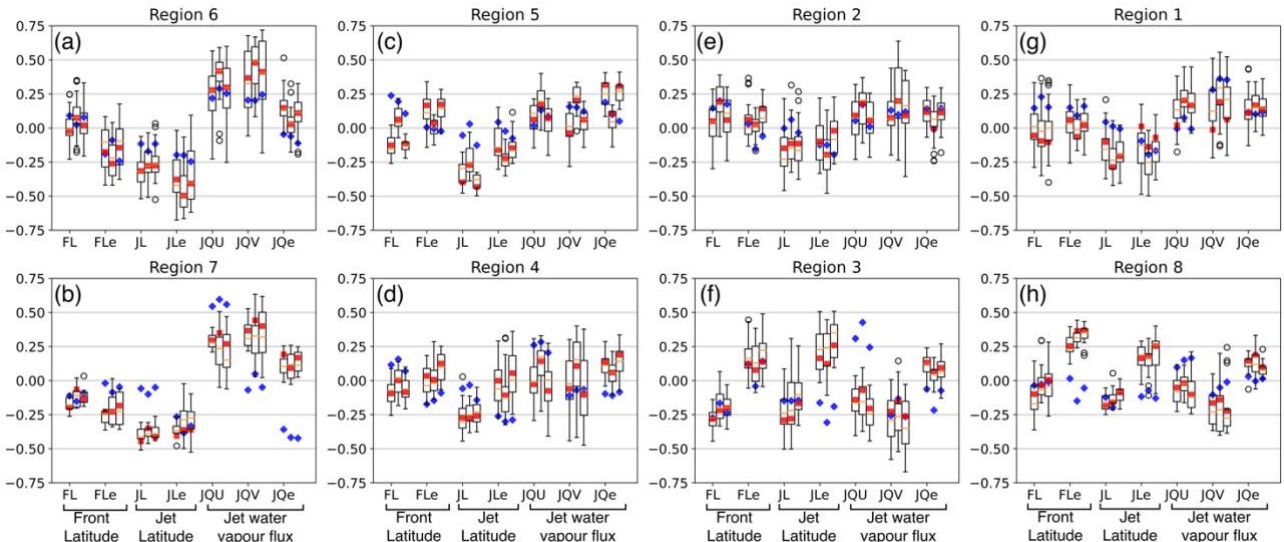


**Figure 19. Like Fig. 10 but with the regional monsoon indices. FL and FLe represent monsoon front latitude indices $\mu_{\text{FLat}}$ and $\sigma^2_{\text{FLat}}$. JL and JLe represent monsoon jet latitude indices $\mu_{\text{JLat}}$ and $\sigma^2_{\text{JLat}}$. JQU, JQV, and JQe represent monsoon jet water vapour flux indices $\mu_{\text{QU}}$, $\mu_{\text{QV}}$, and $\sigma^2_{\text{Q(U,V)}}$.**

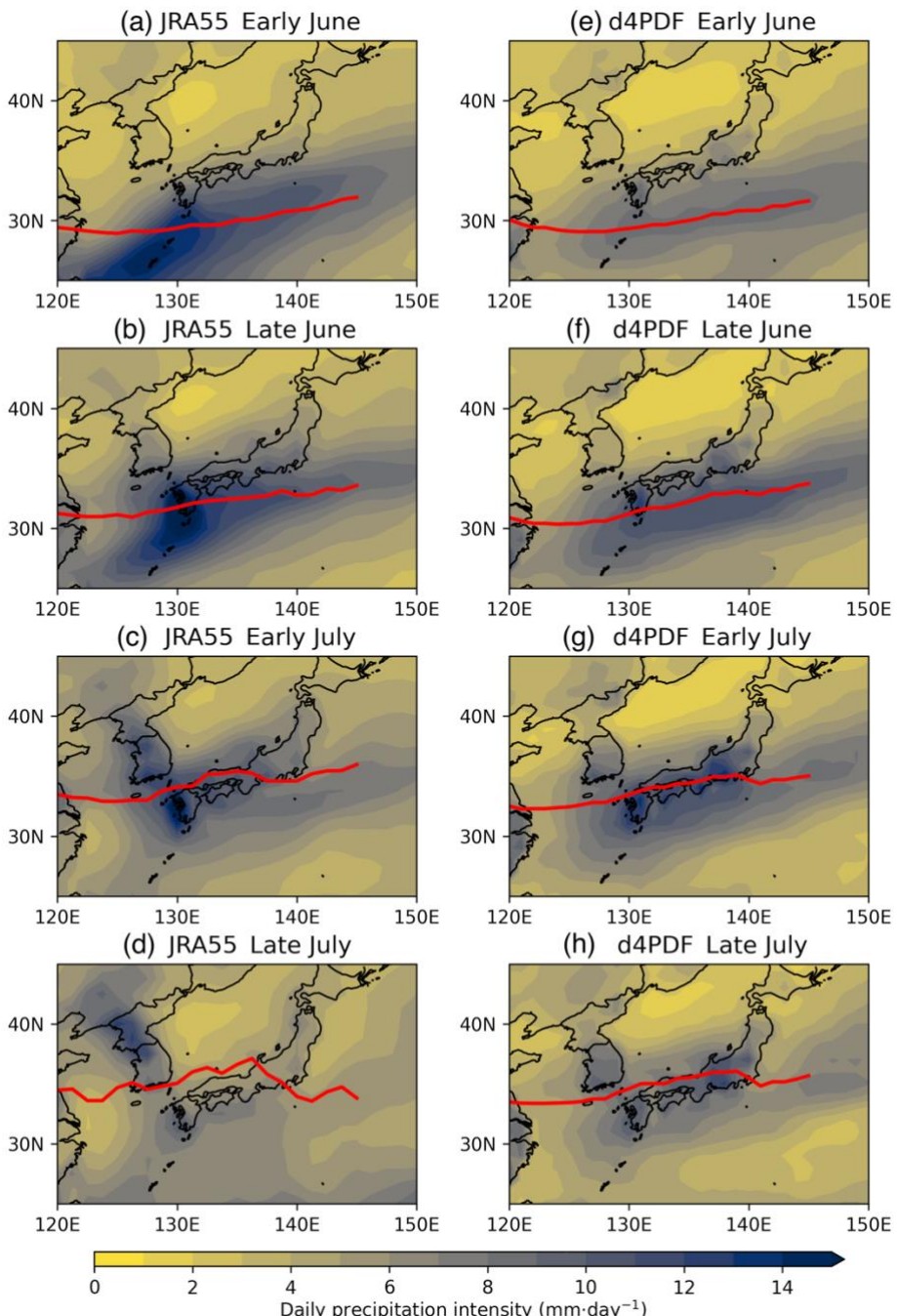


**Figure 19. Daily rainfall in mm day$^{-1}$ (shading) and monsoon front location (red line) for four two-week periods in June–July (rows), comparing JRA55 (left column) and d4PDF (right column). For JRA55, (a) first two weeks of June, (b) second two weeks of June, (c) first two weeks of July, (d) second two weeks of July. For d4PDF, (e) first two weeks of June, (f) second two weeks of June, (g) first two weeks of July, (h) second two weeks of July.**


# 5 Conclusion

This study investigated the relationships between the June–July extreme rainfall over the four largest islands of Japan and the hemispheric-scale climate variability of SST anomalies over the Pacific basin. The 99th percentile hourly, 90th percentile daily, and 90th percentile running-pentad rainfall in the 59-year period of 1952–2010 was compared between d4PDF and rain-gauges (observations). Observed extremes of each temporal resolution were independently clustered using the HDBSCAN algorithm based on their inter-location Spearman correlation. The multi-frequency extremes from the wavelet decomposition of hourly rainfall were then clustered based on a mean metric. Both methods produced similar clusters, and eight analysis regions were identified. The wavelet-method was then applied to cluster d4PDF. Using each d4PDF "set" (cluster or unclassified point) as a spatial sample, Spearman correlation was calculated between the rainfall extremes and the scores of five Pacific SST modes. The modes were obtained by the varimax rotation of extended PCA on Pacific SST anomalies from COBE-SST2, and listed in order of explained variance for 1952–2010, reflected canonical ENSO growth (ENSO+), canonical ENSO decay (ENSO-), warming trend mixed with other climate variability such as the NPGO (Trend+), a non-canonical ENSO that persisted into summer (ENSO-NC), and IPO-dominated Pacific Decadal Variability (PDV)

Observed and d4PDF correlation coefficients ($R$'s) were at most weak ($|R| < 0.50$) over most of Japan. Domain-wide field significance was observed between hourly extremes and Trend+, and between daily extremes and PDV. A group of rain-gauges in northwest Kyushu were significantly related to ENSO+ for daily extremes, consistent with Kawamura et al. (2001) and Jin et al. (2005), which found strong rainfall over Fukuoka after strong La Niña events. In comparison, d4PDF was biased, showing excessive widespread correlation with ENSO-, and excessive anti-correlation with ENSO-NC over eastern Japan. The observed PDV relationship was not reproduced. However, spatial correlation patterns for a smoothed Trend+ was similar between d4PDF and observation. Southern Kyushu showed stronger extremes for all three temporal resolutions.

Rainfall extremes were correlated with regional monsoon indices calculated from JRA55 and d4PDF-AGCM. Based on JRA55 and rain-gauges, $\mu_{QU}$ was the most influential index, with domain-wide field correlation from 36 of 126 rain-gauges showing significant correlation, and up to moderate correlation at southern Kyushu. For both cases, the three most influential indices were the seasonal anomaly of monsoon jet zonal water vapour flux ($\mu_{QU}$), anomaly of monsoon jet meridional water vapour flux ($\mu_{QV}$), and variance of monsoon jet latitude ($\sigma^2_{JLat}$). For both cases, $\mu_{QU}$ was significantly correlated with ENSO-. For d4PDF, $\mu_{QV}$ was significantly correlated with ENSO-, but anti-correlated with ENSO+. These relationships are consistent with Naoi et al. (2020) who found atmospheric rivers to be more frequent in years when El Niño transited into La Niña; An El Niño transitioning into a La Niña may be interpreted as combining the effects of the ENSO- mode with the reversed effects of the ENSO+ mode, hence associated with increased $\mu_{QU}$ (both cases) and $\mu_{QV}$ (d4PDF). The relationships between rainfall extremes and climate variability modes could be explained by the impact of the Pacific SST modes on the regional monsoon, in the form of water vapour flux modulation. The explanation was less successful for the Trend+ mode. The excessive relationships in d4PDF could be explained by the AGCM's slower latitudinal movement of the monsoon.

Since SST anomalies rarely occur as pure modes but as a combination of modes, in a particular year a few climate modes may stack to create a stronger modulating effect on rainfall. Such cases would be exacerbated by the influence of the Trend+ mode in the future warmer climate, specifically the strengthening of hourly extremes (Fig. 5c). The statistical results of this study should be further investigated through sensitivity tests using physics-based numerical weather prediction models. Recently, large ensemble climate prediction datasets such as d4PDF have been widely used in order to extract probabilities of occurrence of extreme weather such as tropical cyclones and heavy rainfalls. Attributing of such extreme events to global warming has been one of the foci in climate and impact assessment studies. In identifying the relationship of extreme events with specific climate modes, special care in evaluating the representation and performance of climate prediction datasets is desirable. This study demonstrates one of the approaches for such evaluations.

## Code availability

All software used to process the data are publicly available from their respective project webpages.

## Data availability

COBE-SST may be downloaded from https://ds.data.jma.go.jp/tcc/tcc/products/elnino/cobesst/cobe-sst.html. The d4PDF data may be downloaded from https://diasjp.net/en/service/d4pdf-data-download. Rainfall from meteorological stations data may be downloaded from the JMA website at https://www.data.jma.go.jp/gmd/risk/obsdl/index.php. SOI and TPI climate indices may be downloaded from the NOAA website https://psl.noaa.gov/data/climateindices/list.

## Author contribution

Conceptualisation and Writing: Lee and Takemi. Investigation: Lee and He. Supervision and Project administration: Takemi.

## Competing interests

The authors declare that they have no conflict of interest.

## Acknowledgements

This work was supported by the MEXT-Program for the advanced studies of climate change projection (SENTAN) Grant Number JPMXD0722678534 and was also supported by the Environment Research and Technology Development Fund Number 2-2303 of the Environmental Restoration and Conservation Agency.

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
