# Peer review of "Verifying the relationships among the variabilities of summer rainfall extremes over Japan in the d4PDF climate ensemble, Pacific sea surface temperature, and monsoon activity"

_EGUsphere, 2024_

## Author Response (AR1)

**Response to Reviewer 1**

This manuscript investigates the interannual variations of the observed precipitation extremes over the western Japan during June and July with the three datasets, i.e., the Radar-AMeDAS during 2006–2022, the 54 rain gauge data during 1952–2022, and the 5km mesh 10-member RCM simulations covering the 59-year period 1952–2010. Differences in the data periods result in an interpretation of the results being difficult and ambiguous. Since the correlation coefficients are discussed and not climate values, data from the same time period should be used.

We thank the reviewer for their useful suggestions. Following the reviewer's recommendation, the same 1952–2010 period is now used for rain-gauges, d4PDF, and principle component analysis of SST. The methodology has been simplified; rain-gauges are clustered, and d4PDF grid-points located at rain-gauges are compared to the rain-gauge clusters. With respect to monsoon indices calculated from JRA55 and d4PDF, the same period of 1958–2010 is used, because JRA55 starts from 1958.

Furthermore, the short 17-year time period of the Radar-AMeDAS DATA is questionable in discussing clustering calculations and inter-annual variations. There is no rationale for comparing Radar-AMeDAS and RCM simulations because the model is forced with the observed SSTs, and therefore should be compared to data between the same periods. However, in this case, the common period is only 5 years, 2006-2010. Therefore, clustering and subsequent analysis should be performed on the rain-gauge data for the period 1952-2010, the same period as the d4PDF. I recommend to re-submit the manuscript. One choice would limit the analysis to an evaluation of model performance by comparing long-term rain-gauge and d4PDF simulations.

Following the reviewer recommendation, Radar-AMeDAS is no longer used, and the analysis is now performed only on long-term rain-gauges and d4PDF. Almost all of the results in the revised manuscript has been re-computed with revised methodology, then re-written.

**Response to Reviewer 2**

This paper investigates relationships between precipitation extremes over western Japan and major SST modes over the Pacific based on observational data, and it tries to explain the relationships through the modulation of monsoon activity. Then, the paper verifies the representation of the obtained relationships in d4PDF data. I found it difficult to evaluate this paper due to several issues listed below during the review. Therefore, I would like to reassess the significance of this paper after the authors have addressed these issues.

We thank the reviewer for his careful reading of the manuscript and his useful suggestions. Almost all the results in the manuscript has been re-computed with revised methodology, then re-written.

First, this paper seems to have incorrect notations in the figure numbers and similar references listed below. Authors should carefully proofread the manuscript before submitting it.

We apologise for the many incorrect notations and references. These errors came about due to past revisions of the manuscript. Following the reviewers major comment #4, most of the supplementary material listed below have been removed. The necessary information is now summarised in the main text.

- Lines 350: Supp. Figs S3-3a -> Supp. Figs S1-3a ?

This material has been removed, since the comparison between COBE-SST and COBE-SST2 is not necessary. The summary of the comparison is now stated in Lines 161–162 of the main text, to describe the robustness of the SST modes. However, a comparison of scores from Principle Component Analysis (PCA) of COBE-SST2 using different time periods has been added to Fig. 1, and described in Lines 312–323.

- Lines 351-355: Similar mistakes as above.

This material has been removed. Please refer to response above.

- Supplementary Material Section 1: Supplementary Figure S2-0 -> Supplementary Figure S1-0 ?

This material has been removed, since the plots are not necessary. The values of explained variances from COBE-SST2 are now stated in Lines 285–290 of the main text.

- Supplementary Material Section 1: Similar mistakes in other figure captions in this section.

The remaining material in this section has been removed, because the comparison between COBE-SST and COBE-SST2 is not necessary.

- The caption of Supplementary Table S2-1: "Meteorological stations used, in three columns, with names in English and Japanese. Years listed with the shaded station are those with insufficient data and not considered." It seems that this explanation does not fully correspond with Table S2-1.

This material has been removed. Since the revised manuscript uses more than double this number of rain-gauges, this table would become very large, but probably not necessary.

- Supplementary Material Section 2: Figure S2-2 -> Supplementary Figure S2-1 ?

This material has been removed, since the revised methodology uses d4PDF grid-points at rain-gauge locations.

- Supplementary Material Section 4: Table S3-1 -> Supplementary Figure S4-1 ?

This material has been removed, since radar-AMeDAS is no longer used.

- Supplementary Material Section 4: Placing Figure S4-5 just below Figure S4-4 would be better.

We agree with the reviewer, but this material has been removed. Since the revised manuscript expands the study domain to the large islands of Japan, figures comparing 20 km and 5 km d4PDF in multiple regions would need to be added. This is a lot of material that is probably not necessary.

In addition, the plotting method used in such figures has been changed (Fig. 4 in the revised manuscript). I think it is hard for readers to see the cluster sizes based on color. The size of the clusters are now directly shown as marker sizes. (This comes at the price that topographical details are not as clear. But since the study domain covers a larger geographical extent, i would be difficult to see anyway.)

- Supplementary Table S6-1a, S6-1b, S6-2a, and S6-2b: What does "ENSO-NC" mean in the rightmost column? There is no explanation of this term in the table caption and the manuscript.

We apologise for the lack of explanation. The revised manuscript now evaluates the first five modes of Pacific SST anomalies. "ENSO-NC" refers to non-canoncial ENSO. Lines 285–294 now discusses this mode in relation to the other modes. It is the 5th mode if calculating PCA for the 1921–2023 period, but the 4th mode for the 1952–2010 period (also the d4PDF period). The supplementary tables have been removed since the revised manuscript now uses a simpler method for comparing between d4PDF and rain-gauges.

- Line 9 in the caption of Supplementary Table S6-2a: 99.9th percentile hourly rainfall -> 99th percentile daily rainfall

This material has been removed, since the revised manuscript now uses a simpler method for comparing d4PDF and rain-gauges.

- Line 174: dJF -> DJF

Since the temporal window of the PCA vector was 15 months, the small letter "d" was originally used to differentiate it from the December of the same year after JJA (June-July-August), "D". This has been changed to D[-1] to indicate that this was the December of the previous year before JJA.

- Line 175: Djf -> DJF

The small letters "j" and "f" were originally used to differentiate it from the January and February of the same year before JJA, "J" and "F". This has been changed to J[+1] and F[+1] to indicate that these were the January and February of the next year after JJA.

Second, I think Table 2 is the most crucial result in this paper; however, I could not understand what type of observational data the presented results are based on and which period they covered. Please note this information in the table caption. In addition, I could not understand the results easily because the results of observation and d4PDF are displayed in layers with complex notations. I would like to ask the authors to present the observations and d4PDF separately.

We agree with the reviewer that the notation was excessively complex. Table 2 now lists the correlation coefficients for rain-gauges and d4PDF separately, in columns side-by-side.

Third, I am concerned about the difference in the periods of the Radar-AMeDAS data and the d4PDF data in interpreting the results. Since the overlapping period between the Radar-AMeDAS data and the d4PDF data is short, it would be better to focus on comparing the ground observation data with the d4PDF data. There is no problem with using the Radar-AMeDAS data as supplementary data. Another choice is that using AMeDAS data would provide more spatially dense observational information since the late 1970s.

Radar-AMeDAS is no longer used. Although there was radar data in early periods, there were format and resolution changes over the years. There seemed to be some numerical artifacts in the early time periods, likely due to sparse radar coverage. I have tried to remove these numerical artifacts using simple methods but the results remain unsatisfactory. In addition, the early data resolution is coarser so some spatial interpolation into corresponding rainfall extremes must be applied. This was why I decided not to use the early radar-AMeDAS.

Fourth, in this paper, many figures and tables are presented in the supplementary section and are cited in the main text. However, I do not believe all these figures and tables are necessary to reach the paper's conclusions. Presenting numerous results with little significance only wastes the reader's time. Please carefully select the figures and tables to be included in the paper. Associated with this point, it seems that the analysis of Ph99.9 and Pd99 has a minor role in this paper, so I think this analysis could be omitted.

We agree with the reviewer. All supplementary material has been removed, and whatever is necessary is inside the main manuscript. The number of monsoon indices has been halved from 14 to 7, since the excess indices do not actually provide more information or improve the conclusions.

Regarding upper percentile rainfall, in the original analysis, the 99th percentile hourly rainfall and 90th percentile rainfall in the latitude-longitude boxes were found to be lower than range of values that were calculated from rain-gauges and cluster. To ensure that the conclusions were consistent for the same numerical range of rainfall values, the 99.9th percentile hourly rainfall was also calculated and used to check the 99th percentile hourly rainfall. The 99th percentile daily rainfall was also calculated and used to check the 90th percentile daily rainfall.

[Other comments]

1. In this paper, the SST mode four is interpreted as the Pacific Decadal Variability mode. Please show a temporal correlation between this mode and a well-known climate index, such as the IPO or PDO index, which would be available on a website.

The correlation between the "PDV" mode with the IPO is now described in Lines 309–311. In addition, the correlation between the "ENSO" modes and the SOI is now described in Lines 301–305.

We have done the correlation for both IPO and PDO indices, and both produced very similar correlation coefficients with the "PDV" mode. We prefer the IPO index to the PDO index for two reasons. Firstly, it describes a domain that is more similar to the one used in this manuscript. Secondly, higher frequencies were filtered, and it is the low frequency signal that we would like to verify. The correlation coefficient of 0.49 with the "PDV" mode is not strong, but is the best match amongst the first 5 modes.

2. Line 533: observations -> JRA55

We have removed this table. "JRA55" is used where the correlation coefficients are given in the text, such as Lines 504, 512, 517, 522. Yhe results described as "observations" only when rain-gauges and JRA55 are correlated (Lines 474–475) .

---

## Referee Report (RR1)

Review of the manuscript "Verifying the relationships among the variabilities of summer rainfall extremes over Japan in the d4PDF climate ensemble, Pacific Sea surface temperature, and monsoon activity" by Shao-Yi Lee, Sicheng He and Tetsuya Takemi.

The authors have hypothesized an eventual relationship between Summer (June-July) monsoon rainfall extremes in Japan (at the hour, day and pentad timescales) and some North-Pacific SST seasonal anomaly modes (ENSO; PDO, pseudo-trend), which has been shown to be non-existent or non-statistically significant. This is concomitant, as well, with unclear or non-existent relationship (according to authors and literature) between ENSO and average monsoon rain in Japan. To represent extremes, the authors have used both rain-gauges and the dynamically downscaled D4PDF dataset. In order to aggregate stations and model grid-points, a rather complex cluster analysis is performed, though not using standard geostatistical techniques. The representativeness of the D4PDF dataset to simulate rainfall extremes is somehow ambiguous. For instance, no systematic comparison is made between the distribution of observed extremes and the distribution of simulated extremes. It was done only indirectly through the comparison between extreme rainfall values and climatic indices.

Authors have performed a lot of work, trying many methodological possibilities (e.g. clustering) to present the inexistence of direct control of the extremes by the analyzed climatic indices.

Despite that non-result, the authors have explored (only at the discussion section) the relationship between rainfall extremes and indices of the summer monsoon (Baiu). Moreover, the modulation of monsson indice statistics (average and standard deviation) by the SST Pacific modes has been studied. Those relationships seem robust, new and interesting.

Giving the above considerations, the manuscript can be publishable after concretizing two main tasks (major points):

1) Rewrite some parts of the main manuscript.
2) Explore further the results linking extremes with monsoon and Pacific indices providing physical reasons.

Below are presented the ordered list of corrections asked:

1) Line 15. Are the modes sorted by decreasing explained variance? Clarify. In line 18 'higher modes' refer to the previous order of modes?

2) Line 17-20. Rewrite the phrase of lines 17-20 in a much clear way by splitting it into two sentences.

3) Line 95 Clarify the simulation period of the '100-member historical-warming (HPB) climate ensemble' as well as the temporal and spatial resolution of the referred GCM.

4) Lines 99-100. No systematic comparison is made between the distribution of observed extremes and the distribution of simulated extremes by the 4DPDF dataset. A synthetic study of the representativeness of 4DPDF extremes shall be made.

5) Line 142 and wherever needed: Change 'Principle Component Analysis' to 'Principal Component Analysis', throughout all the manuscript.

6) Line 144. The linear long trend of SST was not removed to get the anomalous SST. It was obtained farther as a mixed mode (TREND+) or pseudo-trend. Authors could be more direct

by correlating the linear SST trend (for instance averaged throughout all Japan) with extremes. Comment on that.

7) Line 146 Authors have performed extended-PCA (extended Principal Component Analysis) and not a simple PCA. This must be referred explicitly and for clarity, since an extended vector merging 5 delayed (5 trimesters centered in JJA) of spatially distributed values have been taken, from which the covariance and its eigen-decomposition was computed. This is like a MSSA (Multi Singular Spectrum Analysis) with embedding dimension 5 and trimestral sampling.

8) Line 168 The spatial clustering of temporal extremes, particularly the rainfall (e.g. Ma et al. 2020 and references therein) has been studied by different authors, namely by using geostatistical techniques. The reference to those works must be included in the manuscript. The considered metric (distance) to cluster rain-gauges and grid points, used in cluster analysis HDBSCAN depends uniquely on the temporal similarity (Spearman rank correlation) between time-series, being equivalent to the F-madogram. However, for spatially distributed data, a geometrical term, weighing the point-wise distance (e.g. Euclidean) must be added to the statistical distance. The omission of the geometrical term leads to fragmented, topologically complex clusters (even not simply connected, i.e. with 'holes'). This apparently was remedied by adopting ad-hoc clustering rules. The authors are asked to comment on that.

Yingzhao Ma, Mengqian Lu, Cameron Bracken, Haonan Chen, 2020. Spatially coherent clusters of summer precipitation extremes in the Tibetan Plateau: Where is the moisture from?, Atmospheric Research, Volume 237, 104841, ISSN 0169-8095,https://doi.org/10.1016/j.atmosres.2020.104841.

9) Line 175. Authors use the procedure 'percentiles were calculated for each individual ensemble member, then the ensemble mean was taken'. This is an alternative to take quantiles of a super-sample (collecting all sub-samples). Please comment on that.

10) Line 184 Say here the value of CC significant at alpha=0.05.

11) Line 202-205. Authors say: 'At each timestep, the maximum rainfall in each set was selected. For example, in a cluster of three points, this would be the point with the highest rainfall and the other two points would be discarded. This was based on the concept that a set included locations which experienced rainfall from the same event, and the maximum rainfall of that event was sampled by an observer that could "see" the entire location rather than only one specific point.' This procedure seems ad-hoc and not well supported statistically. The maximum of the extremes within a cluster is taken, instead for instance the cluster average of maxima. What is the representativeness of the maximum (among the extremes) within the cluster?

12) Line 220. Authors say: 'Since there were over a hundred rain-gauges, we would like to group them into regions that persisted across the three time resolutions'. The obtained clusters depend on the time resolution (hour, day or pentad), what seems expected, since the associated meteorological systems have different spatio-temporal scales. Authors shall provide a clear justification for their choice. For instance, if instead we had taken the timescales: hour, 6 hours, 3 days, would the clustering output be the same? Ideally, a uniform (or time-scale integrated) criterium should be adopted. Comment on that.

13) Line 220-221. Authors have performed cluster analysis for each of the three analyzed extremes and then they have grouped the respective clusters into common regions. Explain the rationale of that.

14) Line 241 Authors use both the symbol R and CC for the Spearman correlation. Choose only one symbol.

15) Lines 275 The seasonal anomalous Jlat, Flat, QU and QV were obtained as residuals from a sinusoidal and logistic fit of daily values. The result can be biased due to choice of the fit. Why not computing explicitly the daily-basis seasonal cycle from averages along the 61 days with the same Julian day (or within a smoothing window of 2-3 days)?

16) Line 285 Emphasize again that extended-PCA is applied, not PCA, to help the reading. The extended-PCA (Ext-PCA) analysis in the North-Pacific, shown by authors, have omitted other relevant modes of SST Pacific variability like the Northern Pacific Gyre Oscillation (NPGO) (DiLorenzo et al. 2008) that can be also candidates as drivers of extremes. Moreover, the Ext-PCA looks for variability all along the year (all trimesters) and not uniquely focused in the summer season. For instance, a PCA of SST in (JJA) would capture uniquely the difference between summers, eventually producing more suited modes driving the summer rain in Japan. Comment on that.

Di Lorenzo E., Schneider N., Cobb K. M., Chhak, K, Franks P. J. S., Miller A. J., McWilliams J. C., Bograd S. J., Arango H., Curchister E., Powell T. M. and P. Rivere, 2008: North Pacific Gyre Oscillation links ocean climate and ecosystem change. *Geophys. Res. Lett.*, 35, L08607, doi:10.1029/2007GL032838.

17) Line 323 The so-called TREND+ mode seems to contain interannual variability. It is not clear how much is attributed to the trend and global warming. It seems to be a mixed mode including the linear trend (maybe nor the largest part), the PDV (or more commonly the PDO: Pacific Decadal Oscillation) and the NPGO.

18) Line 476. The relationship between monsoon parameters and rainfall extremes is done uniquely for daily extremes. Explain the omission of the other scales.

19) Lines 481-483 Authors say: 'The spatial patterns of correlation were similar between d4PDF and rain-gauges for the $\mu$FLat, $\mu$JLat, and $\mu$QV, showing correlation along the Sea of Japan coast and anti-correlation along the Pacific coast (Figs 8e, 8g, 9e)'. This statement is based on very few rain-gauges, while in some others the agreement is neutral (weak correlations). For other parameters the similarity is quite inexistent, so any apparent similarity seems to be unfair.

20) Line 503. Authors refer to 'scores of the Pacific SST'. Do you mean PCs (ENSO+, ENSO-, TREND+ etc.)? Rewrite and clarify. A Table with correlations between PCs and monsoon indices should be presented with analysis of significance and discussed afterwards. No physical explanation is provided for the correlations found at the end of the Discussion Sec. 4. Improvement needed.

---

## Author Response (AR2)

**Reply to Reviewer 2**

We would like to thank the reviewer for his/her patience and time spent reviewing the manuscript. The methodology has significantly changed in this revision in response to reviewer 3. The results are not qualitatively changed, but we have reduced the number of rain-gauge clusters from 20 to 6. This was done by merging some clusters together using a larger minimum cluster size parameter.

1- Line 16-17: "The correlation patterns for the first two SST modes were somewhat similar between d4PDF and rain-gauges, but were biased spatially and magnitude-wise.": The correlation patterns for the second mode don't look similar between observation and d4PDF (Figs. 5b, 6b, 7b, and Table 2). So, I think the authors need to revise this sentence.

I may be biased psychologically. This sentence has been removed.

2- Line 20: "water vapour flux,." -> "water vapour flux."

The error has been corrected.

3- Line 243: I'm afraid that the threshold of CC is 0.254 in the case of a two-tailed test with the significance level of 0.05 and N=60. Please check it.

We have neglected to state that this was a one-tail test, which has a threshold of 0.22 for N=60 (0.217 for N=59). Since this is not the commonly understood meaning, and $\alpha=0.10$ may be too large for a two-tailed test, we have adjusted the definitions of the correlation (R) strength. The new definitions are "no relation" for $|R| < 0.25$, "weak" for $0.25 \leq |R| < 0.50$, "moderate" for $0.50 \leq |R| \leq 0.75$, "strong" for $0.75 \leq |R| \leq 1.0$, and "perfect" for $|R|=1.0$. The old definition "very strong" was never used. The colormap of the plots have been accordingly adjusted. Figure R2.3.1 below shows how a plot changes from the original colormap (panel c), to one using the two-tailed significance value as the threshold for shading (panel b), to the new definition (panel a). The difference between original and new versions is mainly due to the colour threshold from 0.20 to 0.25. Figures R3.3.2A–R3.3.5B show comparisons of figures in the old manuscript and the new manuscript. New figures are on the left column and old figures are on the right column. Clusters are different from the d4PDF plots because of the revised clustering method, based on Reviewer 3's comments. Also, the mean of clusters is now used instead of the max of clusters, based on Reviewer 3's comments.

[Figure]

**Figure R2.3.1.**

**Panels from Figure 6 of new manuscript**     **Panels from Figure 5 of old manuscript**

[Figure]

Figure R2.3.1A

[Figure]

**Figure R2.3.1B**

**Panels from Figure 7 of new manuscript**     **Panels from Figure 6 of old manuscript**

[Figure]

**Figure R2.3.2A**

**Panels from Figure 7 of new manuscript**  **Panels from Figure 6 of old manuscript**

[Figure]

**Figure R2.3.2B**

[Figure]

**Figure R2.3.3A**

[Figure]

**Figure R2.3.3B**

[Figure]

**Figure R2.3.4A**

**Panels from Figure 14 of new manuscript**  **Panels from Figure 8 of old manuscript**

[Figure]

(e) anomaly with d4PDF  (e) anomaly with d4PDF

(f) variance with d4PDF  (f) variance with d4PDF

Front Latitude

(g) anomaly with d4PDF  (g) anomaly with d4PDF

(h) variance with d4PDF  (h) variance with d4PDF

Jet Latitude

10 50 100 200

-1.0 -0.75 -0.50 -0.25 0.00 0.25 0.50 0.75 1.00

-0.8 -0.6 -0.4 -0.2 0.2 0.4 0.6 0.8

**Figure R2.3.4B**

**Panels from Figure 15 of new manuscript**  **Panels from Figure 9 of old manuscript**

(a) zonal anomaly with individual rain-gauges  (a) zonal anomaly with individual rain-gauges

(b) meridional anomaly with individual rain-gauges  (b) meridional anomaly with individual rain-gauges

Jet water vapour flux

(c) variance with individual rain-gauges  (c) variance with individual rain-gauges

-1.0 -0.75 -0.50 -0.25 0.00 0.25 0.50 0.75 1.00

-0.8 -0.6 -0.4 -0.2 0.2 0.4 0.6 0.8

**Figure S2.3.5A**

[Figure]

**Figure R2.3.5B**

4 - Line 268: For the denominator of the WVflux formula, "pi - pi+1" instead of "pi-l - pi"?

The error has been corrected.

5- Line 279: "For each year, ---" It would be better to insert a line break before this sentence.

A line break has been added.

6- Around Line 505: I think description about the relationship between ENSO+ and monsoon indices should be added here.

Lines 697–716 have been added to discuss ENSO+. (The Discussion section has been reorganised and expanded.)

7- Table 2: There are too many numbers listed, making it difficult to understand the feature intuitively. How about presenting them as a bar graph?

The clustering method has been changed but the clusters produced by the new method are similar to the original manuscript. The value of $c_{min}$ was then increased until smaller clusters merged into 7 clusters. One more cluster was added from $c_{min}=5$ labelled cluster 8. The clusters are plotted in Fig. S2.7.1. The correspondence between the original clusters and the new clusters is shown in Table S2.7.1. The correlation values are now represented as box and whisker plots, Figs 10, 12 and 19 of the new manuscript.

[Figure]

**Figure S2.7.1**

**Table S2.7.1**

| Old cluster | Original | New | New cluster |
|---|---|---|---|
| 1 | Asahikawa, Haboro, Kitami-esashi, Omu, Rumoi, Wakkanai | Asahikawa, Haboro, Kitami-esashi, Omu, Rumoi, Wakkanai, Iwamizawa, Kutchan, Otaru, Sapporo, Suttsu, Tomakomai, Urakawa, Muroran, Obihiro, Esashi, Hakodate, Mutsu, Aomori, Fukaura, Akita, Morioka, Sakata, Kushiro, Miyako, Hachinohe | 1 |
| 2 | Iwamizawa, Kutchan, Otaru, Sapporo, Suttsu | | |
| 3 | Tomakomai, Urakawa | | |
| 4 | Esashi, Hakodate, Mutsu | | |
| 5 | Akita, Morioka, Sakata | | |
| | | | |
| 6 | Fukushima, Ishinomaki, Sendai | Fukushima, Ishinomaki, Sendai, Yamagata, Shirakawa | 8 |
| 7 | Choshi, Katsuura, Mito, Tateno | Choshi, Katsuura, Mito, Tateno, Tokyo, Yokohama, Ajiro, Mishima, Shizuoka, Hamamatsu, Omaezaki, Irozaki, Kumagaya, Okunikko, Chichibu, Kofu, Lake Kawaguchi, | 3 |
| 8 | Tokyo, Yokohama, Kumagaya | | |
| 9 | Chichibu, Kofu, Lake Kawaguchi | | |
| 10 | Fushigi, Kanazawa, Takata, Toyama | Fushigi, Kanazawa, Takata, Toyama, Fukui, Takayama, Wajima, Karuizawa, Suwa, Matsumoto, Nagano | 2 |
| 11 | Fukui, Takayama, Tsuruga | | |
| | | | |
| 12 | Gifu, Hikone, Kyoto, Nagoya | Tsuruga, Iida, Gifu, Hikone, Kyoto, Nagoya, Himeji, Kobe, Nara, Osaka, Tsu, Ueno, Sumoto, Tokushima, Wakayama | 4 |
| 13 | Himeji, Kobe, Nara, Osaka, Tsu, Ueno | | |
| 14 | Sumoto, Tokushima, Wakayama | | |
| 16 | Matsue, Sakai, Yonago | Matsue, Sakai, Yonago, Tottori, Toyooka, Maizuru | 5 |
| 15 | Fukuyama, Hiroshima, Kure, Okayama | Fukuyama, Hiroshima, Kure, Okayama, Tadotsu, Takamatsu, Hagi, Shimonoseki, Fukuoka, Hirado, Hita, Iizuka, Saga, Sasebo, Kumamoto, Mount Unzen, Nagasaki, Akune, Hitoyoshi, Ushibuka, Hamada, Matsuyama, Oita, Uwajima | 6 |
| 17 | Hagi, Shimonoseki | | |
| 28 | Hirado, Hita, Iizuka, Saga, Sasebo | | |
| 29 | Kumamoto, Mount Unzen, Nagasaki | | |
| | | | |
| 20 | Aburatsu, Kagoshima, Makurazaki, Miyazaki, Miyakonojo | Aburatsu, Kagoshima, Makurazaki, Miyazaki, Miyakonojo, Tanegashima, Yakushima | 7 |

**Reply to Reviewer 3**

We would like to thank the reviewer for his/her insightful comments on clustering. Most significantly the metric used for clustering has been revised according to their suggestion, and almost all of the results have been recalculated. The Discussion has also been expanded on.

**Major points**

1) Rewrite some parts of the main manuscript.

The manuscript has been mostly rewritten based on the reviewer's comments.

2) Explore further the results linking extremes with monsoon and Pacific indices providing physical reasons.

The Discussion has been rewritten, to describe how the extremes-monsoon and monsoon-Pacific relationships combine to produce the extremes-Pacific relationships. An explanation is now provided for the difference between model and observation, which is that the model showed a the two week delay in northward movement of the monsoon.

**Corrections**

1) Line 15. Are the modes sorted by decreasing explained variance? Clarify.

The abstract now lists the modes in order of explained variance for the 1952–2010 period of analysis with ENSO-NC listed first then PDV. This is now explicitly stated in L15–17. The order of the 4th and 5th modes differed depending on the analysis period. For longer time periods, the PDV mode accounted for more of the variance than ENSO-NC, but for the 1952–2010 period ENSO-NC accounted for more variance than PDV.

In line 18 'higher modes' refer to the previous order of modes?

The abstract now states the modes explicitly.

2) Line 17-20. Rewrite the phrase of lines 17-20 in a much clear way by splitting it into two sentences.

The sentence has been changed to "correlations between the monsoon indices and standardised principal components of the Pacific SST modes were calculated."

3) Line 95 Clarify the simulation period of the '100-member historical-warming (HPB) climate ensemble' as well as the temporal and spatial resolution of the referred GCM.

The spatial and temporal resolutions of the GCM and RCM are now both stated in L98–99.

4) Lines 99-100. No systematic comparison is made between the distribution of observed extremes and the distribution of simulated extremes by the 4DPDF dataset. A synthetic study of the representativeness of 4DPDF extremes shall be made.

A comparison for 6 analysis regions has been added in Section 3.3, L487–499 and Figure 6. Verification of the d4PDF rainfall extremes was indeed done first before calculating relationships with SST modes. This was inside an earlier version of the manuscript, but then removed at the request of an earlier reviewer. A detailed comparison for 83 regions using radar is currently under review.

5) Line 142 and wherever needed: Change 'Principle Component Analysis' to 'Principal Component Analysis', throughout all the manuscript.

The error has been corrected.

6) Line 144. The linear long trend of SST was not removed to get the anomalous SST. It was obtained farther as a mixed mode (TREND+) or pseudo-trend. Authors could be more direct by correlating the linear SST trend (for instance averaged throughout all Japan) with extremes. Comment on that.

A comparison has been made with a smoothed Trend+ mode in Section 3.3 L550–569 and Figs 11–12. A 11-year moving average had a long enough window (by eye) to sufficiently smooth the scores.

As for why the SSTs were not detrended first, our viewpoint is that results should emerge naturally from the PCA of the data without any assumption of warming (even though we know there is warming). Also, if a linear trend was first calculated, the trend might change depending on the time period chosen. This would make the comparison of different time periods in Section 3.1 more complicated. I think Trend+ reflects the difficulty in distinguishing the warming trend from decadal/multi-decadal variability.

We did assume that the effect of warming on the seasonal cycle was small, when the long-term mean seasonal cycle was removed. The first few attempts at ePCA tried to include the seasonal cycle, but the results were too complicated to work with.

Figures R3.6.1A and R3.6.1B below compare plots of correlation done with smoothed and original Trend+. Left columns show smoothed and right columns show original Trend+. Rows show the temporal resolution of the extremes.

[Figure]

Figure R3.6.1A

Figure R3.6.1B

7) Line 146 Authors have performed extended-PCA (extended Principal Component Analysis) and not a simple PCA. This must be referred explicitly and for clarity, since an extended vector merging 5 delayed (5 trimesters centered in JJA) of spatially distributed values have been taken, from which the covariance and its eigen-decomposition was computed. This is like a MSSA (Multi Singular Spectrum Analysis) with embedding dimension 5 and trimestral sampling.

A statement has been added in L155–156 to clarify this. L157 and L160 now uses "extended PCA". Later mentions of "PCA" in the text have been replaced with "rePCA" (rotated extended Principal Component Analysis).

8) Line 168 The spatial clustering of temporal extremes, particularly the rainfall (e.g. Ma et al. 2020 and references therein) has been studied by different authors, namely by using geostatistical techniques. The reference to those works must be included in the manuscript. The considered metric (distance) to cluster rain-gauges and grid points, used in cluster analysis HDBSCAN depends uniquely on the temporal similarity (Spearman rank correlation) between time-series, being equivalent to the F-madogram. However, for spatially distributed data, a geometrical term, weighing the point-wise distance (e.g. Euclidean) must be added to the statistical distance. The omission of the geometrical term leads to fragmented, topologically complex clusters (even not simply connected, i.e. with 'holes'). This apparently was remedied by adopting ad-hoc clustering rules. The authors are asked to comment on that.

Yingzhao Ma, Mengqian Lu, Cameron Bracken, Haonan Chen, 2020. Spatially coherent clusters of summer precipitation extremes in the Tibetan Plateau: Where is the moisture from?, Atmospheric Research, Volume 237, 104841, ISSN 0169-8095,https://doi.org/10.1016/j.atmosres.2020.104841.

References to previous work have been added in L214–216. The lack of references was not done on purpose and due to ignorance. We came up with the metric with the ad-hoc rules after we intuited that k-means clustering was unsuitable for extremes.

Almost of all the results presented in the manuscript have been recalculated with a new metric that added a geometrical term. It also includes different frequency bands, based on the reviewer's comment 12. Table R3.8.1 below shows clusters from the original manuscript (first column), after adding a geometric term (second column), and using frequency decomposition (third column). The minimum cluster size of $c_{min}$=3 was used. The clusters were similar between the three methods. Increasing $c_{min}$ merged the clusters. Figure R3.8.1 shows the dendrogram. We cut at $c_{min}$=6 and add cluster $f$ from $c_{min}$=5 to get 8 clusters. More comparisons between the old hourly/daily/running-pentad method and the new method are shown the reply to reviewer's comment 12.

**Table R3.8.1.** Bold text indicates changes from the first column, red text indicates changes from second column.

| From merging results of hourly, daily, and running-pentad (Original manuscript) | | Integrated over wavelet decomposition (New) |
|---|---|---|
| Triangulation (original manuscript) | Geometric term added (New) | |
| Asahikawa, Haboro, Kitami-esashi, Omu, Rumoi, Wakkanai | Asahikawa, Haboro, Kitami-esashi, Omu, Rumoi | Asahikawa, Haboro, Kitami-esashi, Omu, Rumoi, Wakkanai |
| Iwamizawa, Kutchan, Otaru, Sapporo, Suttsu | Iwamizawa, Kutchan, Otaru, Sapporo, Suttsu | Iwamizawa, Kutchan, Otaru, Sapporo, Suttsu |
| Tomakomai, Urakawa | | Tomakomai, Urakawa, **Muroran**, Obihiro |
| Esashi, Hakodate, Mutsu | Esashi, Hakodate, **Muroran** | Esashi, Hakodate, Mutsu, Aomori, Fukaura |
| Akita, Morioka, Sakata | Akita, Morioka, Sakata | Akita, Morioka, Sakata |
| Fukushima, Ishinomaki, Sendai | Fukushima, Ishinomaki, Sendai, **Yamagata** | Fukushima, Ishinomaki, Sendai, **Yamagata**, Shirakawa |
| Choshi, Katsuura, Mito, Tateno, Tokyo, Yokohama, Kumagaya | Choshi, Katsuura, Mito, Tateno, Tokyo, Yokohama, Kumagaya | Choshi, Katsuura, Mito, Tateno, Tokyo, Yokohama, |
| | **Ajiro**, **Mishima**, **Shizuoka** | **Ajiro**, **Mishima**, **Shizuoka** |
| | **Hamamatsu**, **Omaezaki** | **Hamamatsu**, **Omaezaki**, Irozaki |
| Chichibu, Kofu, Lake Kawaguchi | *Group exists, but unclassified for daily extremes* | Kumagaya, Okunikko, Chichibu, Kofu, Lake Kawaguchi, |
| | | Karuizawa, Suwa, Matsumoto |
| Fushigi, Kanazawa, Takata, Toyama | Fushigi, Kanazawa, Takata, Toyama | Fushigi, Kanazawa, Takata, Toyama, Fukui, Takayama, Wajima |
| Fukui, Takayama, Tsuruga | | |
| Gifu, Hikone, Kyoto, Nagoya | Gifu, Hikone, Kyoto, Nagoya, | Tsuruga, Iida, Gifu, Hikone, Kyoto, Nagoya, |
| Himeji, Kobe, Nara, Osaka, Tsu, Ueno | Himeji, Kobe, Nara, Osaka, Ueno | Himeji, Kobe, Nara, Osaka, Tsu, Ueno |
| Sumoto, Tokushima, Wakayama | Sumoto, Tokushima, Wakayama | Sumoto, Tokushima, Wakayama |
| Fukuyama, Hiroshima, Kure, Okayama | Fukuyama, Hiroshima, Kure, Okayama | Fukuyama, Hiroshima, Kure, Okayama, Tadotsu, Takamatsu, Hagi |
| Hagi, Shimonoseki | Hagi, Shimonoseki, | Shimonoseki, Fukuoka, Iizuka |
| Hirado, Hita, Iizuka, Saga, Sasebo | Hita, Iizuka, Saga, Sasebo | Hirado, Hita, Saga, Sasebo |
| Matsue, Sakai, Yonago | Matsue, Sakai, Yonago, **Tottori** | Matsue, Sakai, Yonago, **Tottori**, Toyooka, Maizuru |
| Kumamoto, Mount Unzen, Nagasaki | Kumamoto, Mount Unzen, Nagasaki | Kumamoto, Mount Unzen, Nagasaki |
| Aburatsu, Kagoshima, Makurazaki, Miyazaki, Miyakonojo | Aburatsu, Kagoshima Makurazaki, Miyazaki, Miyakonojo | Aburatsu, Kagoshima, Makurazaki, Miyazaki, Miyakonojo, Tanegashima, Yakushima |
| | | Akune, Hitoyoshi, Ushibuka |

[Figure]

**Figure R.3.8.1**

9) Line 175. Authors use the procedure 'percentiles were calculated for each individual ensemble member, then the ensemble mean was taken'. This is an alternative to take quantiles of a super-sample (collecting all sub-samples). Please comment on that.

During the verification of climatological extremes (for the whole 59 year period) we have compared taking ensemble mean and taking super-sample. The results were similar and in that case we felt the super-sample method was better, with percentiles of individual ensemble members forming the range of uncertainty about the super-sample value.

In this study a percentile is calculated for each year. Each year has its own SST and each ensemble member a perturbation of the SST, but we wish to do a correlation between the "real" SST interannual-variability and the model results. When a percentile super-sample is taken, the ensemble members do not participate equally. We have the concern that one or a few ensemble members may dominate the tail, but they do not reflect a response to the "real" SST. Instead they could reflect the response to a subset of the perturbed SSTs, and this subset could change every year. Then extreme rainfall time series could reflect the interannual variation of subsets, rather than the interannual variation of the real SST. To guard against such a possibility, we take an ensemble mean of percentiles, to force equal participation by each ensemble member. This is different from the climate case above, where the interannual variation is included in the samples.

10) Line 184 Say here the value of CC significant at alpha=0.05.

The threshold values are now explicitly stated in L210–212 and L276–278.

11) Line 202-205. Authors say: 'At each timestep, the maximum rainfall in each set was selected. For example, in a cluster of three points, this would be the point with the highest rainfall and the other two points would be discarded. This was based on the concept that a set included locations which experienced rainfall from the same event, and the maximum rainfall of that event was sampled by an observer that could "see" the entire location rather than only one specific point.' This procedure seems ad-hoc and not well supported statistically. The maximum of the extremes within a cluster is taken, instead for instance the cluster average of maxima. What is the representativeness of the maximum (among the extremes) within the cluster?

Actually, I have calculated the maximum, mean, median and the 99th/90th percentiles of the clusters. But my statistical knowledge is quite poor and I could not decide which was best to use. The time series of the statistic is correlated against other time series (SST mode scores, monsoon indices). Also, multiple clusters from d4PDF correspond to one rain-gauge cluster, so a further statistic had to be calculated from many clusters. I did not understand how to interpret such calculations when performed on a statistic, so I decided to just pick the point with the maximum value in each cluster. In my mind the downstream calculations then reduced to the typical interpretation. As the reviewer points out, this can produce a time series unrepresentative of the cluster, since the points inside each cluster were not perfectly correlated in time. We have switch to the cluster mean for both rain-gauges and d4PDF. Figures R3.11.1A– R3.11.2B show comparison of plots in the original manuscript using cluster mean (left column) and using cluster max (old column). The results are qualitatively similar, but the correlation values of large clusters are affected, such as the very big cluster in north Hokkaido (most extreme right side of the panels).

[Figure]

**Figure R3.11.1A.** d4PDF correlation between scores of Pacific SST modes 99th percentile hourly extremes.

[Figure]

**Figure R3.11.1B.** d4PDF correlation between scores of Pacific SST modes 90th percentile daily extremes.

[Figure]

**Figure R3.11.1C.** d4PDF correlation between scores of Pacific SST modes and 90th percentile running-pentad extremes.

[Figure]

**Figure R3.11.2A.** d4PDF correlation between regional monsoon indices related to latitude and 90th percentile daily extremes.

[Figure]

**Figure R3.11.2B.** d4PDF correlation between regional monsoon indices related to water vapour flux and 90th percentile daily extremes.

12) Line 220. Authors say: 'Since there were over a hundred rain-gauges, we would like to group them into regions that persisted across the three time resolutions'. The obtained clusters depend on the time resolution (hour, day or pentad), what seems expected, since the associated meteorological systems have different spatio-temporal scales. Authors shall provide a clear justification for their choice. For instance, if instead we had taken the time-scales: hour, 6 hours, 3 days, would the clustering output be the same? Ideally, a uniform (or time-scale integrated) criterium should be adopted. Comment on that.

A multi-frequency metric has been defined. The hourly rainfall time series was decomposed into different frequency components using wavelets. (Due to the highly intermittent nature of hourly rainfall, the Fourier decomposed components looked unreasonable. In contrast, wavelet decomposed components showed matching peaks with the original time series.) Figure R3.12.1 shows a comparison of $c_{min}=6$ groups formed using the old method (panel a) and using the new metric (panel b). Cluster 8 in panel b was manually added. Table R3.12.1 shows how the clusters are similar. The new method is better because it covers the whole frequency range without choosing certain frequency band from a priori meteorological knowledge. Also, this allows for the clusters in d4PDF to be the same between the hourly, daily and running-pentad analysis.

Unfortunately, the problem of many fragmented clusters still occurs for d4PDF. Preliminary investigations using single frequency bands or single ensemble member suggest that this may be due to a combination of high frequency component (also associated with systems of small spatial scale) and multiple ensemble members (each responding to a different perturbed SST). I am unable to solve the problem at this point in time.

[Figure]

**Figure R3.12.1**

13) Line 220-221. Authors have performed cluster analysis for each of the three analyzed extremes and then they have grouped the respective clusters into common regions. Explain the rationale of that.

We wished to use the same set of clusters when comparing between correlations for hourly, daily, and running-pentad extremes. The methodology has changed into a single cluster analysis with a multi-frequency metric (response to reviewer's comment 12).

14) Line 241 Authors use both the symbol R and CC for the Spearman correlation. Choose only one symbol.

Only R is now used.

15) Lines 275 The seasonal anomalous Jlat, Flat, QU and QV were obtained as residuals from a sinusoidal and logistic fit of daily values. The result can be biased due to choice of the fit. Why not computing explicitly the daily-basis seasonal cycle from averages along the 61 days with the same Julian day (or within a smoothing window of 2-3 days)?

The smoothing window was our first method of choice. Figure R3.15.1 shows our attempts at smoothing the monsoon-related variable for JRA55 (left column) and d4PDF (right column). We did not feel satisfied with the results of even a 15-day window, so we did fits instead.

[Figure]

Figure R3.15.1

16) Line 285 Emphasize again that extended-PCA is applied, not PCA, to help the reading. The extended-PCA (Ext-PCA) analysis in the North-Pacific, shown by authors, have omitted other relevant modes of SST Pacific variability like the Northern Pacific Gyre Oscillation (NPGO) (DiLorenzo et al. 2008) that can be also candidates as drivers of extremes. Moreover, the Ext-PCA looks for variability all along the year (all trimesters) and not uniquely focused in the summer season. For instance, a PCA of SST in (JJA) would capture uniquely the difference between summers, eventually producing more suited modes driving the summer rain in Japan. Comment on that.

Di Lorenzo E., Schneider N., Cobb K. M., Chhak, K, Franks P. J. S., Miller A. J., McWilliams J. C., Bograd S. J., Arango H., Curchister E., Powell T. M. and P. Rivere, 2008: North Pacific Gyre Oscillation links ocean climate and ecosystem change. *Geophys. Res. Lett.*, 35, L08607, doi:10.1029/2007GL032838.

"PCA" in the text have been changed to "rePCA" (rotated extended Principal Component Analysis). A comparison of the SST modes with NPGO has been added in Section 3.1 L351–253 (reviewer's comment 17). Rainfall extremes have been correlated with NPGO, TPI, SOI in Section 3.3 L565–567 and Figure 11. The reason for extended PCA was specifically to select for all-year variability, and clarify the timescale of the influential phenomena. Climate indices are calculated by month, but the SST climate mode are slow, such as ENSO developing and decaying over two years. A monthly decomposition may include both the direct anomaly of the slow mode, and the indirect anomaly due to the summer (monsoon) response to the mode. If it is primarily the summer response, different climate modes may project onto similar regional SST (or atmospheric) anomalies that drive a similar monsoon response. This seems to be the case based on the results of this study, so indeed analysis of regional summer markers should produce more suitable modes.

17) Line 323 The so-called TREND+ mode seems to contain interannual variability. It is not clear how much is attributed to the trend and global warming. It seems to be a mixed mode including the linear trend (maybe nor the largest part), the PDV (or more commonly the PDO: Pacific Decadal Oscillation) and the NPGO.

A comparison of the SST modes with NPGO has been added in Section 3.1 L364–366. The NPGO was significantly correlated with the Trend+ mode with $R=0.38$. This is the best $R$ value amongst the five modes. In order to isolate the response to warming alone, Section 3.3 L550–569 and Figs 11–12 now has additional analysis using Trend+ smoothed by an 11-year moving average.

18) Line 476. The relationship between monsoon parameters and rainfall extremes is done uniquely for daily extremes. Explain the omission of the other scales.

Figures have been added for hourly extremes as Figs 13–14, for running-pentad extremes as Figs 17–18, and summarised regionally for the three resolutions as Fig. 19. These figures were left out in the previous manuscript because I thought there would be too many figures, but they are unavoidable to properly discuss the results.

19) Lines 481-483 Authors say: 'The spatial patterns of correlation were similar between d4PDF and rain-gauges for the $\mu$FLat, $\mu$JLat, and $\mu$QV, showing correlation along the Sea of Japan coast and anti-correlation along the Pacific coast (Figs 8e, 8g, 9e)'. This statement is based on very few rain-gauges, while in some others the agreement is neutral (weak correlations). For other parameters the similarity is quite inexistent, so any apparent similarity seems to be unfair.

I may have made some unfair claims on similarity due to psychological bias. To prevent this, the manuscript now focuses on rain-gauges using objective criteria such as statistical significance within the rain-gauge cluster and field significance for rain-gauges or clusters. To compare d4PDF against rain-gauges, the box-and-whisker plots for the 8 analysis regions are used, such as Figs 6, 10, 12, and 19. The Discussion now focuses on explaining how the regional monsoon acts as an intermediary between the Pacific SST modes and rainfall extremes, for each case of rain-gauges and d4PDF.

20) Line 503. Authors refer to 'scores of the Pacific SST'. Do you mean PCs (ENSO+, ENSO-, TREND+ etc.)? Rewrite and clarify. A Table with correlations between PCs and monsoon indices should be presented with analysis of significance and discussed afterwards. No physical explanation is provided for the correlations found at the end of the Discussion Sec. 4. Improvement needed.

The sentence has been rewritten as "correlations between the monsoon indices and standardised principal components of the Pacific SST modes were calculated". Table 2 has been added showing the correlations. Figure 20 has been added to show the bi-weekly progression of the monsoon in JRA55 and d4PDF. We offer the explanation that differences between JRA55 and d4PDF was due to the slower northward movement of the monsoon in d4PDF.